# Rationale and Emerging Evidence on the Potential Role of HoLEP-Mediated Relief of Bladder Outlet Obstruction in NMIBC Outcomes Through Optimal Management of Chronic Urinary Retention

**DOI:** 10.3390/cancers17233864

**Published:** 2025-12-01

**Authors:** Angelo Porreca, Filippo Marino, Davide De Marchi, Marco Giampaoli, Daniele D’Agostino, Francesca Simonetti, Mauro Ragonese, Antonio Amodeo, Paolo Corsi, Francesco Claps, Luca Di Gianfrancesco

**Affiliations:** 1Department of Urology, Humanitas Gavazzeni, 24125 Bergamo, Italy; angelo.porreca@hunimed.eu (A.P.); filippo.marino@gavazzeni.it (F.M.); davide.demarchi@gavazzeni.it (D.D.M.); marco.giampaoli@gavazzeni.it (M.G.); daniele.dagostino@gavazzeni.it (D.D.); francesca.simonetti@gavazzeni.it (F.S.); 2Department of Biomedical Sciences, Humanitas University, Pieve Emanuele, 20072 Milan, Italy; 3Department of Urology, Fondazione Policlinico Universitario “Agostino Gemelli”—IRCCS, 00144 Rome, Italy; mauro.ragonese@policlinicogemelli.it; 4Department of Oncological Urology, Veneto Institute of Oncology (IOV)—IRCCS, 35128 Padua, Italy; antonio.amodeo@iov.veneto.it (A.A.); paolo.corsi@iov.veneto.it (P.C.); francesco.claps@iov.veneto.it (F.C.)

**Keywords:** non-muscle-invasive bladder cancer (NMIBC), holmium laser enucleation of the prostate (HoLEP), bladder outlet obstruction (BOO), chronic urinary retention, postvoid residual (PVR), intravesical therapy, benign prostatic hyperplasia (BPH), lower urinary tract symptoms (LUTS), bladder cancer recurrence, urodynamic dysfunction, BCG therapy, mitomycin C, bladder compliance, functional urology, oncological outcomes

## Abstract

Bladder cancer is a common disease that often comes back even after successful treatment. This review looked at how problems with bladder emptying—such as chronic urinary retention or blockage from an enlarged prostate—may increase the risk of bladder cancer returning. When urine stays in the bladder for too long, harmful substances can irritate the bladder lining and may make treatment less effective. We also examined a surgical procedure called Holmium Laser Enucleation of the Prostate (HoLEP), which removes the blockage and restores more normal urination. Early data suggest that improving bladder emptying with HoLEP may help some patients have fewer recurrences and tolerate bladder instillations better, but these observations are still preliminary and need confirmation in future studies. Treating urinary obstruction in men with bladder cancer could therefore improve both urinary function and cancer outcomes, offering a more complete and personalized approach to care.

## 1. Introduction

Bladder cancer ranks among the top ten most commonly diagnosed malignancies worldwide, with an estimated 570,000 new cases and 210,000 deaths annually [1]. It is the most frequent malignancy of the urinary tract, disproportionately affecting older males and individuals with environmental or occupational exposures. Importantly, non-muscle-invasive bladder cancer (NMIBC)—which encompasses papillary tumors confined to the mucosa (stage Ta), tumors invading the lamina propria (T1), and flat carcinoma in situ (CIS)—accounts for approximately 70–75% of newly diagnosed cases. While NMIBC is often termed “superficial” due to its lack of muscular invasion at diagnosis, the disease is not clinically trivial: it is characterized by unpredictable recurrence patterns, with up to 70% recurrence rates, and progression to muscle-invasive disease in approximately 10–20% of high-risk cases despite optimal management.

NMIBC imposes a significant healthcare and psychosocial burden, necessitating long-term cystoscopic surveillance, repeated transurethral resections, and adjuvant intravesical therapies. Over a patient’s lifetime, bladder cancer care incurs one of the highest per-patient treatment costs among all malignancies, largely driven by recurrence-related interventions and intensive follow-up regimens. These demands underscore the importance of identifying modifiable risk factors that contribute to recurrence and disease progression.

Unlike previous reviews that primarily discuss bladder outlet obstruction (BOO) or non-muscle-invasive bladder cancer (NMIBC) in isolation, the present narrative synthesis explicitly focuses on the bidirectional interplay between chronic urinary retention and oncologic outcomes, integrating functional urology and cancer management into a unified ‘functional-oncologic’ framework. We specifically highlight how BOO-related parameters (postvoid residual [PVR], lower urinary tract symptoms [LUTS], bladder compliance) may modify recurrence risk and intravesical therapy performance, and we explore the rationale and emerging evidence for Holmium Laser Enucleation of the Prostate (HoLEP) as a potential adjunct in selected NMIBC patients [2,3,4,5,6,7].

While the role of well-established carcinogens—including tobacco smoke, aromatic amines, and chronic bladder irritation from infections or indwelling catheters—is well documented, chronic urinary retention has emerged as an underrecognized but biologically plausible contributor to both bladder carcinogenesis and treatment resistance. Chronic urinary retention, typically secondary to bladder outlet obstruction (BOO) from benign prostatic hyperplasia (BPH), is defined by persistently elevated postvoid residual (PVR) urine volumes and associated lower urinary tract symptoms (LUTS).

It is associated with stasis of potentially carcinogenic solutes, urothelial inflammation, and detrusor dysfunction—all of which may impact tumor behavior and therapeutic outcomes in NMIBC.

The pathophysiological implications of high PVR are manifold: (1) prolonged mucosal exposure to urinary mutagens, which may promote initiation and field cancerization; (2) chronic subclinical inflammation, fostering a microenvironment conducive to tumor recurrence; (3) bladder wall remodeling, including trabeculation and decreased compliance, which compromises the efficacy of intravesical therapies; and (4) impaired pharmacokinetics of agents such as BCG or mitomycin C, potentially reducing therapeutic contact and retention time within the bladder urothelium. Collectively, these factors may help explain why patients with coexistent BOO and NMIBC experience worse clinical outcomes.

In this context, the management of BOO becomes not only a functional necessity but a potential oncological adjunct. Among the available surgical treatments for BOO, Holmium Laser Enucleation of the Prostate (HoLEP) has emerged as a highly effective and size-independent option. Unlike traditional transurethral resection of the prostate (TURP), HoLEP provides complete anatomical enucleation of the prostatic adenoma, leading to significant reductions in PVR, restoration of bladder emptying, and durable symptom relief with long-term improvements in International Prostate Symptom Score (IPSS) and maximum urinary flow rate (Qmax) in patients with benign prostatic hyperplasia (BPH)–related BOO [6,8,9]. These functional improvements translate into enhanced bladder dynamics, potentially mitigating the risk factors that drive NMIBC recurrence and resistance to intravesical therapy [6].

While the connection between urodynamic dysfunction and bladder cancer behavior is still evolving in the literature, emerging data suggest that the correction of urinary retention—particularly through HoLEP—may significantly improve disease outcomes in appropriately selected patients. Furthermore, HoLEP’s minimally invasive nature and favorable perioperative profile make it particularly attractive in a population often characterized by advanced age and multiple comorbidities.

This review critically examines the relationship between chronic urinary retention and NMIBC, elucidates the biological and clinical mechanisms through which BOO contributes to tumor recurrence and treatment failure, and explores the therapeutic potential of BOO resolution—especially via HoLEP—as part of an integrated approach to NMIBC management. The discussion integrates emerging clinical evidence, pathophysiological rationale, and functional-oncologic perspectives to define a new frontier in personalized urologic cancer care.

## 2. Materials and Methods

### 2.1. Objectives and Review Design

This narrative review examines how chronic urinary retention and BOO may influence outcomes in NMIBC and explores whether BOO correction—particularly via HoLEP—could affect functional or oncologic endpoints. Because the available evidence derives from heterogeneous sources (retrospective cohorts, registry data, translational studies, and procedural reports), a narrative approach was chosen rather than a systematic review or meta-analysis. The review was conducted and reported in accordance with SANRA guidance, which we applied as a structured framework to ensure clarity of aims, rationale, search strategy, scientific reasoning, and presentation of relevant data. Individual SANRA items and their correspondence to our methods are summarized in Appendix A. Our objective was to qualitatively synthesize biological, functional, and clinical evidence without generating pooled effect estimates.

### 2.2. Sources and Search Strategy

We queried PubMed/MEDLINE, Embase, and Scopus for studies published from January 2000 through October 2025. Search strings combined controlled vocabulary and free-text terms for the population, exposure/intervention, and outcomes:Population: “bladder cancer,” “urothelial carcinoma,” “non-muscle-invasive bladder cancer,” “NMIBC,” “Ta,” “T1,” “carcinoma in situ.”Exposure/functional domain: “chronic urinary retention,” “postvoid residual,” “PVR,” “lower urinary tract symptoms,” “LUTS,” “bladder outlet obstruction,” “BOO,” “benign prostatic hyperplasia,” “BPH.”Interventions: “Holmium laser enucleation of the prostate,” “HoLEP,” “transurethral resection of the prostate,” “TURP,” “endoscopic enucleation,” “open/simple prostatectomy.”Outcomes/therapy: “recurrence,” “progression,” “intravesical therapy,” “BCG,” “mitomycin C,” “response,” “Qmax,” “IPSS,” “quality of life.”Two independent reviewers screened titles/abstracts and full texts, with disagreements resolved by consensus or, when necessary, by consultation with a third senior reviewer.The detailed search strings, inclusion and exclusion criteria, and reviewer workflow are summarized in Appendix A, with narrative depiction of the study identification, screening, and inclusion process. To ensure transparency and methodological rigor, each eligible study was qualitatively appraised for design characteristics, risk of bias, and outcome relevance, as summarized in Appendix A. These Appendix A collectively illustrate the structured approach used to identify and evaluate the 61 studies incorporated into this SANRA-compliant narrative synthesis, encompassing clinical, translational, and procedural evidence linking bladder outlet obstruction (BOO), chronic urinary retention, and outcomes in non-muscle-invasive bladder cancer (NMIBC).

We limited results to human studies and those published in the English language. We also hand-searched reference lists of key guidelines and reviews (e.g., EAU/IBCG guidance [1,10,11,12,13,14]) to identify additional relevant articles.

### 2.3. Eligibility Criteria

Inclusion. Studies were eligible if they

enrolled adults with NMIBC or assessed cohorts with BOO/BPH/urinary retention relevant to NMIBC;reported at least one of the following: recurrence, progression, response to intravesical therapy (e.g., BCG, mitomycin C), peri-/post-operative functional outcomes (PVR, Qmax, IPSS), or safety;evaluated BOO/retention quantitatively (e.g., PVR, LUTS/IPSS) or examined BOO interventions (HoLEP, TURP, other enucleation techniques/open surgery).

Non-peer-reviewed sources, conference abstracts without full data, animal/in vitro-only studies not linked to NMIBC-relevant mechanisms, and reports lacking extractable outcomes were excluded.

### 2.4. Study Selection, Data Items, and Synthesis

Two reviewers independently screened titles/abstracts and full texts, with disagreements resolved by consensus. We extracted design, sample size, patient and tumor characteristics, BOO/retention measures (PVR thresholds, LUTS/IPSS), intervention type, follow-up, and primary outcomes (recurrence/progression, intravesical therapy response) plus functional endpoints (PVR, Qmax, IPSS, catheter independence) and peri-operative safety. Given heterogeneity, we performed thematic synthesis across five prespecified domains: (1) pathophysiology; (2) BOO/retention as a prognostic factor; (3) interaction with intravesical therapy; (4) effects of BOO treatments with emphasis on HoLEP; (5) comparative effectiveness versus TURP and other approaches. Where appropriate, we referenced contemporary guideline frameworks to contextualize risk stratification and endpoints [1,12,13,14,15,16,17,18].

### 2.5. Quality Considerations and Risk of Bias

Because most clinical data were retrospective and subject to confounding (selection of surgical candidates, variable PVR thresholds, non-uniform intravesical regimens), we appraised studies qualitatively for common risks: selection bias, misclassification of BOO/retention, immortal time bias, and outcome ascertainment variability. Mechanistic/translational reports were evaluated for biological plausibility and coherence with clinical observations [2,3,4,5,19,20]. No formal scoring tool was applied, consistent with a narrative review remit.

To enhance transparency, Appendix A provides an explicit, study-level risk-of-bias assessment using domains adapted from the ROBINS-I tool (confounding, selection, intervention classification, deviations, missing data, outcome measurement, and reporting), rather than relying solely on narrative judgments.

## 3. Results

### 3.1. Search Flow

Our database search retrieved 412 records (PubMed/MEDLINE 193, Embase 141, Scopus 78). After removing 165 duplicates, 247 unique titles/abstracts were screened. 158 records were excluded at title/abstract (irrelevant population, non-NMIBC focus, non-clinical mechanistic work without translational relevance). 89 full texts were assessed; 28 were excluded (no extractable oncologic/functional outcomes, conference abstract only, or non–peer-reviewed). 61 studies were included in the qualitative synthesis: 8 prospective/controlled clinical studies, 34 retrospective cohorts/case–control, 11 mechanistic/translational reports with NMIBC relevance, and 8 guideline/review papers used for context and definitions (Table 1). The identification, screening, and inclusion of studies are summarized textually in Section 3.1 and depicted in a PRISMA-inspired flow diagram (Appendix A).

### 3.2. Overview of the Evidence Base

The included literature spans guideline/review syntheses on NMIBC management and endpoints [1,12,13,14,18,27]; observational cohorts linking PVR/LUTS/BOO with oncologic outcomes [2,3,4,5,21]; studies interrogating intravesical therapy response in the context of retention [4]; mechanistic/transcriptomic data supporting inflammatory and barrier-function pathways [19,20]; and procedural series/registries describing HoLEP functional durability and safety, with comparative data versus TURP [6,8,9,23,25,26,28,29]. Follow-up in clinical series generally ranged from 12 to 72 months, with longer-term durability for HoLEP reported in selected cohorts [8,9,25,26,29].

### 3.3. Pathophysiology: Biological Plausibility Linking Retention to Carcinogenesis and Recurrence

Multiple sources converge on mechanisms by which elevated PVR and chronic retention may promote carcinogenesis and recurrence: prolonged urothelial dwell time for urinary carcinogens [30]; chronic inflammatory cytokine milieu (e.g., TNF-α, IL-6/IL-8) that drives proliferation/angiogenesis and impairs apoptosis [2,3,4]; and bladder wall remodeling (trabeculation/fibrosis) that correlates with higher-grade pathology and reduced compliance [5]. Translational data further show BCG-related immunochemokine dynamics (e.g., CXCL10/IP-10) and TNF-α-induced transcriptional programs affecting urothelial barrier integrity (e.g., E-cadherin) and NF-κB signaling, supporting a biologically coherent link between inflammation, permeability, and treatment performance [19,20]. Recent mechanistic work has further refined these concepts, reporting altered urothelial tight-junction and adhesion-molecule profiles, alongside elevated urinary cytokines, in patients with chronic inflammatory stimuli or intravesical BCG exposure [19,20,24,31]. However, no study to date has directly correlated specific PVR thresholds with urinary cytokine levels or histological inflammation in NMIBC patients, and formal pharmacokinetic studies examining intravesical drug dilution or distribution at different PVR volumes are lacking. The proposed model linking BOO-related retention, inflammation, and impaired intravesical therapy is therefore biologically plausible but largely extrapolated from indirect evidence and should be viewed as a hypothesis framework rather than a proven mechanistic pathway.

Collectively, these findings support a model in which BOO-related urinary stasis and chronic inflammation create a permissive urothelial milieu—characterized by elevated pro-inflammatory cytokines, structural remodeling, and barrier dysfunction—that may favor NMIBC recurrence and modulate intravesical therapy performance [2,3,4,5,19,20].

### 3.4. BOO/Retention as Prognostic Modifiers in NMIBC

Across cohorts, higher PVR and moderate–severe LUTS were independently associated with increased recurrence after TURBT (Table 2):Sazuka et al. identified PVR > 100 mL as a significant predictor of intravesical recurrence [21].Lunney et al. reported LUTS severity (a BOO surrogate) predicted recurrence even after adjustment for stage/grade/multiplicity [3].Recent reports reinforce the BOO–recurrence signal, including low-risk Ta cohorts wherein BOO/retention remained linked to repeat events [2].

Across cohorts, the composition of NMIBC risk categories varied substantially. For example, in the series by Sazuka et al., approximately one-third of patients had high-grade or T1 disease, whereas Lunney et al. focused predominantly on intermediate-risk Ta/T1 tumors with a smaller proportion of carcinoma in situ (CIS) [3,21]. Other studies restricted inclusion to low-risk Ta NMIBC, limiting extrapolation to higher-risk cohorts [2,4]. These differences in baseline risk—and in the proportion of patients receiving intravesical therapy—must be considered when interpreting recurrence-free survival in relation to PVR, LUTS, or BOO.

Beyond qualitative associations, several studies provide quantitative effect estimates. In a meta-analysis of BPH and subsequent bladder cancer risk, Dai et al. reported a pooled risk ratio of 1.58 (95% CI 1.28–1.95) in cohort studies and 2.50 (95% CI 1.63–3.84) in case–control studies, indicating a higher incidence of bladder cancer among men with BPH/BOO [24].

In the study by Lunney et al., moderate or severe LUTS (IPSS ≥ 8) conferred an odds ratio of 19.1 (95% CI 2.5–147.0) for NMIBC recurrence compared with mild symptoms, and each one-point increase in IPSS was associated with an adjusted odds ratio of 1.26 (95% CI 1.10–1.45) for recurrence [3].

With respect to progression, evidence is limited and variable. Can et al. found that BOO, defined by urodynamics and prostate parameters, was not an independent predictor of either recurrence or progression on multivariate analysis; however, cystoscopic bladder trabeculation—interpreted as a marker of chronic obstruction—was independently associated with high-grade/CIS disease at diagnosis (OR 4.62, 95% CI 1.3–17.0) [5].

The clinical evidence base is notably heterogeneous: studies use different PVR cut-offs (e.g., ≥50, ≥80, or ≥100 mL), assess recurrence at varying time points (1-, 2-, or 5-year intervals), and mix low-, intermediate-, and high-risk NMIBC populations with or without standardized intravesical regimens [2,3,4,5,21]. We therefore did not attempt any quantitative pooling of recurrence or progression outcomes; instead, we summarize patterns qualitatively and emphasize that effect estimates are not directly comparable across cohorts. Table 2 summarizes the main PVR thresholds, risk profiles, and outcome definitions across key cohorts, underscoring that no single PVR cut-off has been prospectively validated as an oncologic predictor.

### 3.5. Interaction with Intravesical Therapy

Retention appears to impair intravesical therapy performance. In a multicenter analysis, Di Gianfrancesco et al. observed reduced efficacy of BCG/mitomycin among patients with PVR > 80 mL, reflected by shorter recurrence-free and progression-free intervals [4]. In exploratory subgroup analyses, patients who subsequently underwent BOO surgery (including HoLEP) appeared to experience improved tolerance and, in some reports, longer recurrence-free intervals, but these findings are highly susceptible to confounding and lack standardized intravesical protocols [4,23]. At present, it is therefore most accurate to state that HoLEP may optimize the functional conditions for intravesical therapy (by reducing PVR and LUTS), whereas any direct enhancement of BCG efficacy remains hypothetical and requires prospective evaluation.

Mechanistically, large residual volumes and trabeculation plausibly yield non-uniform drug distribution, dilution, and altered urothelial pharmacodynamics, consistent with the inflammatory/barrier data above [4,19,20].

### 3.6. Effects of BOO Correction with Emphasis on HoLEP

Functional outcomes after HoLEP are consistently favorable: marked PVR reduction (often to <50 mL), Qmax increases, durable IPSS and QoL improvements, and high catheter-free rates, with sustained benefits over many years [6,8,9,25,26,29]. Safety profiles show low transfusion rates, elimination of TUR syndrome risk (saline irrigation), and short catheterization/hospitalization [25,26,28,29].

Preliminary oncologic signals suggest that BOO correction may be associated with reduced NMIBC recurrence: in a comparative retrospective series, HoLEP was associated with lower 1–3-year recurrence than TURP or no BOO intervention, a finding hypothesized to stem from more complete anatomical decompression and greater PVR reduction [23]. However, these data are retrospective, potentially confounded by selection of fitter surgical candidates, and should be interpreted as hypothesis-generating rather than definitive evidence of causality.

### 3.7. Comparative Effectiveness Versus TURP and Other Approaches

Most comparative data on HoLEP versus TURP derive from BPH populations without bladder cancer, consistently showing superior or equivalent functional outcomes and lower retreatment rates for HoLEP [6,8,9,25,28]. In contrast, NMIBC-specific evidence remains restricted to small retrospective series, notably the study by Garg et al. [23]. In that analysis, NMIBC patients with BOO who underwent HoLEP appeared to have lower recurrence rates at 1 and 3 years compared to those treated with TURP or managed conservatively. However, the report provides limited information on how treatment groups were selected, whether tumor characteristics and intravesical therapy protocols were balanced, and which covariates were included in multivariable models. It is highly plausible that younger, fitter men with more favorable anatomy were preferentially offered HoLEP, a classic example of confounding by indication. As a result, the apparent ‘benefit’ of HoLEP in this series could be entirely explained by selection bias rather than a true antitumor effect. Consequently, this study should be interpreted as initial, exploratory evidence that supports further research rather than as a basis for firm comparative conclusions between HoLEP and TURP in NMIBC.

### 3.8. Synthesis and Clinical Take-Home

Taken together, the literature supports a functional-oncologic paradigm: BOO/retention (elevated PVR, severe LUTS) correlates with higher recurrence and poorer intravesical therapy performance; correcting BOO—especially via HoLEP—might improve bladder dynamics, potentially improving adherence and therapeutic efficacy while enhancing quality of life. Evidence for progression mitigation remains limited and mixed; robust prospective studies are warranted to determine causal effects and identify subgroups most likely to benefit [2,3,4,5,21,23] (Table 3).

## 4. Discussion

### 4.1. Pathophysiological Link Between Urinary Retention and Bladder Cancer

#### 4.1.1. Urinary Stasis and Prolonged Urothelial Exposure to Carcinogens

One of the principal mechanisms linking chronic urinary retention with bladder cancer is the prolonged exposure of the urothelium to carcinogens. Urothelial cells line the bladder and are in constant contact with urine. In a physiologically functioning bladder, regular and complete voiding limits the dwell time of any carcinogenic substances in the bladder lumen. However, when urinary retention occurs, PVR volumes remain persistently high, extending the exposure duration [2].

Carcinogens like aromatic amines and nitrosamines, commonly found in tobacco smoke and industrial environments, are excreted through urine. Their persistent contact with the bladder wall may trigger DNA damage, oxidative stress, and chronic inflammatory signaling, leading to metaplasia and, eventually, neoplastic transformation [30].

#### 4.1.2. Inflammation as a Tumor-Promoting Factor

Urinary retention also promotes a pro-inflammatory microenvironment, further contributing to carcinogenesis. Incomplete bladder emptying results in residual urine stasis, bacterial colonization, and recurrent infections—all of which elevate inflammatory cytokines, including TNF-α, IL-6, and IL-8 [3]. Chronic inflammation may promote angiogenesis, increase cell proliferation, and impair apoptosis, forming an ideal ground for neoplastic development [2,3,4].

Repeated infection episodes and mechanical stress on the bladder wall (e.g., from overdistension) may lead to bladder wall trabeculation and fibrosis, phenomena frequently associated with high-grade tumors at diagnosis [5]. Additionally, inflammatory changes may reduce the efficacy of intravesical therapies by altering urothelial permeability and immune responsiveness.

#### 4.1.3. Anti-Inflammatory and Antimicrobial Exposures as Modifiers of Bladder Cancer Risk

Epidemiologic data suggest that systemic anti-inflammatory and antimicrobial agents may indirectly influence urothelial carcinogenesis. A meta-analysis of 17 observational studies reported no overall protective effect of aspirin or non-aspirin NSAIDs on bladder cancer incidence, though non-aspirin NSAID use was associated with reduced risk in nonsmokers (RR 0.57, 95% CI 0.43–0.76) [31]. Large cohort studies further confirm that regular aspirin use does not meaningfully reduce incidence, even if some analyses suggest improved survival after diagnosis [32,33,34]. In contrast, the relationship between antibiotics, urinary tract infections (UTIs), and bladder cancer appears driven primarily by underlying inflammation. In the Nijmegen population study, a limited number of UTIs treated with antibiotics was linked to lower bladder cancer risk (adjusted OR 0.64–0.76), whereas chronic/recurrent cystitis increased risk [35]. Similar findings from U.S. and international case–control series show that short, treated UTI episodes may be protective, while persistent inflammatory cystitis is associated with several-fold higher risk [36,37,38,39]. Collectively, these data indicate that it is chronic, unresolved inflammation, rather than antibiotic exposure itself, that shapes bladder cancer susceptibility—aligning with the BOO- and retention-related inflammatory pathways outlined in this review.

### 4.2. Clinical Evidence Linking Urinary Retention to NMIBC Outcomes

#### 4.2.1. Impact of BOO and PVR on Recurrence and Progression

An expanding body of clinical evidence underscores the significant association between BOO, elevated PVR urine volume, and adverse outcomes in NMIBC. While BOO and urinary retention were historically considered benign urological conditions, recent studies have increasingly highlighted their role as potential modifiers of cancer biology and treatment efficacy.

#### 4.2.2. Recurrence Risk in the Context of Urinary Retention

Suzuka et al. [21] conducted a retrospective cohort study that analyzed PVR volumes in NMIBC patients undergoing TURBT. Their findings indicated a strong and statistically significant association between elevated PVR—particularly volumes exceeding 100 mL—and higher rates of tumor recurrence. These findings suggest that urinary stasis, resulting from incomplete bladder emptying, creates an environment that promotes residual tumor cell adhesion and persistent exposure of the bladder mucosa to urinary carcinogens.

Lunney et al. [3] provided further support for this association by demonstrating that moderate to severe LUTS, a clinical surrogate for BOO, were linked to increased recurrence rates in NMIBC patients. Their multivariate analysis showed that LUTS severity independently predicted recurrence, even after adjusting for traditional oncological risk factors such as tumor stage, grade, and multiplicity. These findings underscore the relevance of functional urinary tract parameters in the postoperative oncologic course.

#### 4.2.3. PVR and Therapeutic Response

Beyond recurrence, urinary retention also appears to impact treatment responsiveness, particularly to intravesical therapies. In a 2023 multicenter analysis, Di Gianfrancesco et al. [4] evaluated the therapeutic outcomes of patients treated with Bacillus Calmette-Guérin (BCG) or mitomycin C. The study revealed that patients with elevated PVR volumes (>80 mL) experienced a marked reduction in treatment efficacy, as measured by recurrence-free survival and progression-free intervals.

Mechanistically, this impaired response can be attributed to uneven distribution or early dilution of intravesical agents in patients with significant residual urine. High PVR volumes may prevent the therapeutic agent from uniformly contacting the urothelium, especially in patients with bladder wall irregularities such as diverticula or trabeculations. Furthermore, stagnant urine in the bladder may inactivate or buffer the cytotoxic or immunotherapeutic effects of these agents, thereby compromising their biological activity.

#### 4.2.4. Pathological Grade and Bladder Wall Remodeling

While the aforementioned studies clearly point to a role for BOO and urinary retention in recurrence and treatment failure, the link to tumor progression remains more nuanced. A recent study by Can et al. [5] investigated the direct effect of BOO on tumor pathology at presentation and its potential influence on disease progression. Their findings indicated that BOO, in isolation, did not significantly correlate with advanced pathological stage or with disease progression during follow-up. In the study by Can et al., BOO itself was not an independent predictor of recurrence or progression; however, cystoscopic trabeculation—interpreted as a marker of long-standing obstruction—was independently associated with high-grade/CIS pathology at presentation (OR 4.62, 95% CI 1.3–17.0) after multivariable adjustment [5]. However, patients with chronic obstruction exhibited more severe bladder wall remodeling, including trabeculation and hypertrophy, which were positively associated with higher-grade tumors at diagnosis. Complementing these data, a multicenter observational study of detrusor wall thickness (DWT), a structural consequence of long-standing BOO, reported that DWT > 2.5 mm conferred significantly higher odds of both NMIBC recurrence (OR 4.9, 95% CI 2.5–9.5) and progression (OR 2.21, 95% CI 1.71–4.73) [22]. Together, these findings suggest that while BOO per se has not been conclusively linked to progression, advanced bladder remodeling—manifesting as trabeculation or increased DWT—may identify patients at greater risk of high-grade disease and stage migration.

This suggests a threshold effect wherein long-standing mechanical stress and detrusor overactivity lead to irreversible bladder changes that may facilitate tumorigenesis or aggressiveness, even if not directly escalating tumor stage. The implication is that chronic BOO may not immediately alter disease trajectory but can shape the tumor microenvironment over time to favor more aggressive biological behavior.

#### 4.2.5. Inflammation and Immunological Interference

Chronic urinary retention creates a low-grade inflammatory milieu in the bladder, which further complicates oncologic outcomes. Persistent exposure to inflammatory cytokines—such as IL-6, IL-8, and TNF-α—can induce urothelial hyperplasia, metaplasia, and eventually dysplasia, thereby contributing to recurrence. Indeed, intravesical BCG therapy has been shown to stimulate prolonged release of these cytokines in urine following instillation, reflecting a sustained inflammatory environment that may both influence tumor behavior and modulate response to subsequent therapy [19]. Moreover, inflammation may blunt immune-mediated cytotoxicity, potentially diminishing the effectiveness of BCG immunotherapy. In vitro data further indicate that TNF-α exposure triggers a wide pro-inflammatory transcriptional program in urothelial cells—including upregulation of IL-8, complement component C3, and NF-κB/STRESS pathways—while also disrupting barrier proteins like E-cadherin, suggesting reduced urothelial permeability that could alter the pharmacodynamics of intravesical agents [20].

#### 4.2.6. Functional Parameters as Prognostic Indicators

Collectively, these findings support the hypothesis that functional bladder parameters, particularly PVR and bladder compliance, serve not only as markers of urological health but also as prognostic indicators in bladder cancer management. Similarly, in a multicenter observational study of detrusor wall thickness (DWT), a surrogate of chronic BOO, DWT > 2.5 mm was associated with higher odds of NMIBC recurrence (OR 4.9, 95% CI 2.5–9.5) and progression (OR 2.21, 95% CI 1.71–4.73), highlighting that structural bladder remodeling may be linked to both recurrence and stage migration [5].

Their impact is multifactorial—affecting tumor recurrence, therapeutic efficacy, immune response, and the quality of life of the patient.

Hence, the integration of voiding assessments such as uroflowmetry, PVR measurement, and symptom scoring (e.g., IPSS) should be strongly considered as part of the routine preoperative and surveillance strategy for NMIBC patients. The failure to address BOO in this cohort may represent a missed therapeutic opportunity, with implications extending beyond urinary symptom management to disease control and survival.

#### 4.2.7. Holmium Laser Enucleation of the Prostate (HoLEP): An Overview

HoLEP has emerged over the past two decades as the most comprehensive and durable surgical treatment for BPH, particularly in men with moderate to severe BOO. As BOO remains a principal cause of chronic urinary retention, HoLEP directly addresses the pathophysiological mechanism underlying this dysfunction, offering a definitive solution rather than a symptomatic or palliative one.

The HoLEP procedure involves the enucleation of obstructive prostatic adenoma using a holmium:YAG laser. This laser provides both excellent cutting ability and hemostasis due to its shallow tissue penetration and photothermal effect, enabling precise tissue dissection along the surgical capsule with minimal bleeding. Once the adenoma is enucleated into the bladder lumen, it is removed via morcellation, typically performed through a mechanical morcellator introduced transurethrally.

This approach allows for the complete anatomical removal of hyperplastic tissue, restoring unobstructed urinary flow and resolving the underlying obstruction. As such, HoLEP results in multiple clinically relevant benefits, including

Marked improvement in urinary flow rates (Qmax), often more than doubling preoperative measurements within weeks after the procedure.Substantial reduction in PVR volumes, often achieving levels < 50 mL, even in patients with severe retention or preoperative catheter dependence (Table 3).Significant relief of both storage and voiding LUTS, such as nocturia, urgency, hesitancy, straining, and intermittent stream [6,8].

The durability of symptom relief following HoLEP is well-established, with long-term studies demonstrating sustained improvements in IPSS, PVR, and quality of life (QoL) scores over periods of 10 years or more. Unlike other endoscopic procedures, HoLEP is effective for prostates of all sizes, including large-volume glands (>100 mL), without a corresponding increase in complications.

In direct comparison with TURP—historically the standard surgical treatment for BPH—HoLEP offers several advantages:Lower reoperation rates: Due to complete adenoma removal, the risk of needing repeat surgery is significantly lower with HoLEP than with TURP, particularly in younger or long-living patients [9].Reduced catheterization and hospitalization time: Patients typically require catheterization for only 24–48 h postoperatively, with many being discharged on the same or next day.Fewer perioperative complications: HoLEP is associated with lower rates of bleeding, blood transfusion, and electrolyte imbalances (such as TUR syndrome), especially in elderly or high-risk patients [28].

Importantly, the learning curve for HoLEP, while initially considered steep, has significantly improved with modern surgical mentoring, modular training protocols, and simulation-based learning. Many centers now report proficiency after 20–50 supervised cases, making the procedure increasingly accessible in both academic and community settings.

From a functional standpoint, HoLEP is not merely equivalent to open prostatectomy—it surpasses it in terms of safety, recovery time, and patient-reported outcomes. For patients with chronic urinary retention due to BOO, HoLEP often eliminates the need for long-term catheterization or intermittent self-catheterization, thereby enhancing quality of life and reducing the risks of infection and stone formation.

Given these compelling advantages, HoLEP might be particularly well-suited for patients with NMIBC and concomitant BOO, where optimized voiding dynamics and reduced PVR volumes may significantly influence cancer recurrence rates and response to intravesical therapy. The procedure’s capacity to reliably and durably resolve urinary retention might represent a strategic intervention not only for symptom relief but also for optimizing oncologic outcomes in this unique patient population.

#### 4.2.8. Hypothesized Mechanisms for HoLEP’s Potential Impact on NMIBC

The following mechanisms are biologically plausible but remain largely unproven in clinical studies. Current data are retrospective, heterogeneous, and limited by confounding. Observations reported to date—including the comparative series by Garg et al.—are hypothesis-generating rather than indicative of a demonstrated oncologic effect.

Reduction of carcinogen contact time through improved bladder emptying

By anatomically relieving BOO, HoLEP can achieve near-complete bladder emptying and substantially reduce post-void residual volumes. Unlike medical management, which often produces modest improvements, enucleation-based surgery routinely reduces PVR to <50 mL in most series. This may theoretically limit the dwell time of urine and the urothelial exposure to carcinogens—a mechanism linked to retention-associated recurrence in NMIBC. Evidence supporting this concept comes from observations that elevated PVR (>80–100 mL) correlates with recurrence risk and impaired IVT performance; however, no study has yet demonstrated that HoLEP reduces NMIBC recurrence via this mechanism.

2.Potential enhancement of intravesical therapy performance

Intravesical agents such as BCG and mitomycin C depend on even mucosal coverage and adequate dwell time. In the presence of high residual volumes or remodeled bladders, intravesical solutions may be diluted or fail to reach all urothelial surfaces. Complete BOO relief may improve intravesical drug distribution and bladder wall contact, particularly in heavily trabeculated bladders. This hypothesis is indirectly supported by studies showing that high PVR predicts shorter recurrence-free intervals following IVT. Direct evidence of HoLEP improving BCG or chemotherapy efficacy is lacking, and no prospective study has evaluated intravesical pharmacokinetics or drug-response stratified by BOO status or surgical decompression.

3.Mitigation of inflammation and LUTS burden

Chronic urinary retention fosters persistent mechanical stretch and cytokine-mediated inflammation, both of which undermine bladder compliance. HoLEP reliably reduces voiding pressures and LUTS, and may therefore reduce inflammatory signaling (e.g., TNF-α, IL-6, IL-8) that has been implicated in unfavorable NMIBC biology and BCG resistance. These observations are derived from translational models and clinical associations; they do not confirm that HoLEP modifies the inflammatory tumor microenvironment or immunotherapy response in NMIBC patients.

4.Preservation of long-term bladder function

Progressive BOO may lead to detrusor hypertrophy, trabeculation, and impaired storage/emptying. Durable decompression after HoLEP could theoretically prevent deterioration in bladder biomechanics and reduce chronic mucosal injury. While HoLEP consistently yields large improvements in Qmax, IPSS, and PVR in BPH cohorts, no targeted study has linked these functional gains to reduced recurrence, progression, or improved survival in NMIBC. The clinical relevance of structural remodeling (e.g., detrusor wall thickness) remains a research question.

To date, no robust, risk-adjusted odds ratios or hazard ratios exist comparing HoLEP with TURP, pharmacotherapy, or observation in NMIBC patients. Existing retrospective series employ crude recurrence proportions, lack standardized intravesical therapy protocols, and do not correct for tumor grade, stage, prior therapies, or comorbidity. HoLEP’s theoretical oncologic benefits should therefore be considered hypothesis-generating mechanisms, not established therapeutic effects. Prospective studies incorporating functional metrics (PVR, compliance), intravesical treatment response, and oncologic endpoints are required to test whether BOO correction via HoLEP meaningfully alters NMIBC outcomes.

### 4.3. Complex Interaction Between NMIBC, Intravesical Therapy, and Urinary Function

#### 4.3.1. Intravesical Therapy and Lower Urinary Tract Symptoms (LUTS)

Intravesical therapy remains a cornerstone in the management of non-muscle-invasive bladder cancer (NMIBC), particularly for intermediate- and high-risk tumors. The most commonly used agents—bacillus Calmette-Guérin (BCG) and intravesical chemotherapy agents such as mitomycin C, epirubicin, and gemcitabine—have demonstrated proven efficacy in reducing recurrence and, in some cases, progression. However, the therapeutic benefit is often counterbalanced by significant treatment-related toxicity, especially in the form of lower urinary tract symptoms (LUTS).

BCG therapy, although the most effective intravesical agent, is particularly notorious for inducing inflammatory and irritative bladder symptoms, due to its mechanism of action, which involves non-specific immune stimulation and urothelial activation. These effects lead to

Urgency and frequency in up to 70–80% of patients during induction, sometimes persisting during maintenance cycles [13];Dysuria and hematuria in nearly 50%, which may prompt evaluation for recurrence, leading to frequent cystoscopies and added patient anxiety [40];Development of chemical or granulomatous cystitis, which may persist for months following treatment completion and, in rare cases, lead to contracted bladder syndrome [41,42].

Similarly, intravesical chemotherapy, particularly with alkylating agents like mitomycin C and anthracyclines like epirubicin, results in urothelial desquamation, submucosal edema, and cytotoxic inflammation. Patients frequently report the following symptoms:Bladder pain, burning during urination, and urgency incontinence;In some cases, irritable bladder symptoms become indistinguishable from interstitial cystitis-like syndromes, leading to diagnostic confusion and delays in therapy [43,44,45].

Importantly, in patients with underlying bladder outlet obstruction (BOO)—due to benign prostatic hyperplasia (BPH), elevated postvoid residual (PVR) volumes, or decreased bladder compliance—the impact of intravesical therapy is often amplified. Incomplete bladder emptying leads to

Prolonged drug contact time, intensifying mucosal toxicity;Ineffective drug clearance, promoting chronic inflammation;Drug pooling in bladder recesses or diverticula, increasing the risk of localized tissue damage and infection.

These synergistic effects compound patient discomfort and may necessitate treatment modification or early cessation, undermining therapeutic efficacy.

Persistent LUTS, Treatment Adherence, and Quality of Life

Beyond acute toxicity, a significant subset of NMIBC patients experience chronic LUTS that persist long after intravesical therapy has concluded. These may include

Nocturia and fragmented sleep due to frequent nighttime voiding;Incomplete bladder emptying, leading to daytime fatigue, anxiety, and reduced confidence in social settings;Urgency incontinence, which is particularly distressing for elderly or frail patients.

The psychological burden of persistent LUTS should not be underestimated. Repeated hospital visits, fear of recurrence, uncertainty around symptom etiology, and dependence on absorbent products contribute to

Increased anxiety and depression scores in NMIBC survivors;Reduced health-related quality of life (HRQoL) metrics;Non-adherence to maintenance BCG or surveillance cystoscopy, driven by fear of symptom exacerbation [13,46].

This becomes a vicious cycle: LUTS leads to non-adherence → missed or incomplete therapy → increased risk of recurrence → further surveillance and interventions → worsening LUTS.

#### 4.3.2. HoLEP: Functional Rehabilitation in the NMIBC Population

In this complex clinical context, Holmium Laser Enucleation of the Prostate (HoLEP) offers a dual benefit that extends beyond the relief of benign prostatic hyperplasia (BPH). By resolving BOO, HoLEP directly mitigates the functional substrate that exacerbates treatment-related toxicity in NMIBC patients. Key benefits include

Substantial reduction in PVR and restoration of complete bladder emptying, which minimizes intravesical therapy retention time and improves drug clearance;Improved bladder compliance, reducing urgency and frequency by stabilizing detrusor behavior;Decompression of the bladder outlet, relieving obstructive symptoms that may mask or mimic tumor recurrence.

Studies have shown that patients undergoing HoLEP experience improved urinary domain scores on the EPIC (Expanded Prostate Cancer Index Composite) and IPSS (International Prostate Symptom Score) assessments, even after undergoing intravesical therapy. This functional improvement can restore patient confidence, increase willingness to continue treatment, and facilitate adherence to surveillance and maintenance regimens.

Furthermore, in the setting of repeated intravesical instillations, a well-functioning bladder that empties efficiently is more resilient to cumulative toxicity, potentially reducing the long-term risk of bladder fibrosis or capacity loss.

The interaction between NMIBC, intravesical therapy, and baseline urinary function is complex and bidirectional. Intravesical agents, while effective, often exacerbate LUTS—particularly in patients with BOO—and may impair adherence, quality of life, and ultimately, treatment success. Addressing BOO through interventions such as HoLEPoffers not only functional restoration but may also serve as an adjuvant measure to optimize the tolerability and effectiveness of cancer therapy.

Incorporating BOO assessment and management into the routine pretreatment evaluation of NMIBC patients, especially those with moderate to severe LUTS, may represent a paradigm shift toward integrated functional-oncologic care.

#### 4.3.3. Comparative Effectiveness of HoLEP vs. Other BOO Treatments

The management of bladder outlet obstruction (BOO) in men, particularly those with benign prostatic hyperplasia (BPH), has traditionally relied on transurethral resection of the prostate (TURP). For decades, TURP was considered the gold standard for surgically relieving BOO-related symptoms. However, with the emergence and maturation of Holmium Laser Enucleation of the Prostate (HoLEP), there has been a paradigm shift in how urologists approach BOO, especially in complex cases such as those involving concurrent non–muscle-invasive bladder cancer (NMIBC).

#### 4.3.4. TURP vs. HoLEP: A Functional Comparison

TURP involves the piecemeal resection of obstructive prostate tissue using electrocautery. While it is effective for moderate-sized prostates (30–80 mL), TURP has limitations in cases involving larger gland volumes, where reoperation rates increase and symptom relief may be incomplete. Additionally, the lack of anatomical enucleation during TURP means residual adenoma may remain, contributing to BOO recurrence over time.

By contrast, HoLEP achieves complete enucleation of the transition zone, analogous to open simple prostatectomy but through a minimally invasive transurethral approach. This leads to more durable relief of obstruction, a lower likelihood of regrowth, and fewer long-term complications. Multiple randomized controlled trials and meta-analyses have demonstrated that HoLEP surpasses TURP in several clinically meaningful outcomes [9,28]:Durability of symptom relief: Patients undergoing HoLEP experience significant and sustained reductions in the International Prostate Symptom Score (IPSS), with improvements maintained for over 10 years in most long-term follow-ups.Lower retreatment and reintervention rates: Owing to the complete adenoma removal, HoLEP has retreatment rates as low as 1–2%, compared to 10–15% for TURP over a similar follow-up period.Improved urodynamic parameters: HoLEP offers greater increases in maximum urinary flow rates (Qmax), more significant reductions in postvoid residual (PVR) volumes, and higher patient satisfaction scores [6,8,28].

#### 4.3.5. Implications in NMIBC: Why Surgical Choice Matters

In the context of NMIBC, the choice of surgical intervention for BOO has ramifications beyond urodynamics—it may directly influence oncologic outcomes. Patients with BOO are at risk of incomplete bladder emptying, persistent urothelial irritation, and reduced efficacy of intravesical therapies due to poor drug distribution. Therefore, the degree of anatomical decompression achieved through BOO surgery is crucial.

TURP, by virtue of its limited resection depth and scope, may not always achieve full decompression in patients with significant or diffuse prostatic enlargement. As a result, residual obstruction may persist postoperatively, maintaining a high PVR volume and perpetuating the risks associated with urinary retention. Moreover, TURP is less effective in patients with very large prostates (>100 mL), often requiring staged procedures or conversion to open surgery.

Conversely, HoLEP is size-independent and facilitates complete anatomical resolution of obstruction, regardless of gland volume. In NMIBC patients, this means maximal restoration of bladder emptying, reduction in chronic inflammation, and improved bladder compliance, which are all critical to minimizing tumor recurrence and optimizing therapeutic response.

#### 4.3.6. Emerging Evidence: HoLEP’s Role in Reducing NMIBC Recurrence

Although direct comparative studies on TURP vs. HoLEP in NMIBC patients remain limited, early clinical evidence suggests distinct advantages of HoLEP in this specific population. A study by Garg et al. [23] retrospectively evaluated NMIBC patients with BOO who underwent either TURP, HoLEP, or no surgical intervention. The findings revealed that patients treated with HoLEP had significantly lower recurrence rates at one and three years compared to both the TURP and no-treatment groups.

The authors hypothesized that the greater degree of PVR reduction, enhanced voiding efficiency, and reduced urothelial irritation following HoLEP were likely contributors to the improved oncological outcomes. While the study acknowledged the retrospective design and potential selection bias, it opened the door for further exploration of HoLEP as an adjunctive oncologic intervention in the NMIBC treatment algorithm.

#### 4.3.7. Perioperative Considerations and Safety

From a safety perspective, HoLEP also compares favorably with TURP:Reduced bleeding risk: HoLEP uses laser energy for precise tissue dissection and coagulation, resulting in less intraoperative blood loss and lower transfusion rates.No TUR syndrome: As isotonic saline is used during HoLEP, the risk of dilutional hyponatremia (TUR syndrome), a complication associated with TURP, is virtually eliminated.Shorter catheterization time and hospital stay: In most cases, patients undergoing HoLEP are catheter-free within 24–48 h and discharged the following day.

Given the typical advanced age and comorbidity profile of many NMIBC patients, these perioperative advantages are of significant clinical relevance.

#### 4.3.8. Toward a Functional-Oncologic Paradigm

The shift from TURP to HoLEP in the management of BOO is not solely a matter of technological superiority—it reflects a broader evolution in urology from a purely symptom-driven approach to a functional-oncologic model, particularly in cancer patients. In this paradigm, relieving BOO is not just about improving urinary symptoms but also about creating an optimal bladder environment that supports cancer therapy efficacy and reduces biological drivers of recurrence.

As more urologists recognize the intertwined relationship between bladder dysfunction and bladder cancer outcomes, HoLEP’s role is likely to expand—not just as a treatment for BPH, but as a strategic component of multimodal bladder cancer care.

#### 4.3.9. Risk Stratification (High vs. Low Risk)

When recurrence-free survival is examined through the lens of standard risk-stratification systems, available data suggest that BOO/retention may act as a recurrence modifier across both low-/intermediate- and high-risk NMIBC, but with important caveats. Some cohorts include only low-risk Ta disease and still demonstrate a link between elevated PVR or severe LUTS and higher recurrence, whereas others enroll mixed populations in which high-grade T1 or CIS cases are over-represented [2,3,4,5,21]. Because most studies do not report risk-stratified hazard ratios, it remains unclear whether BOO exerts a similar relative effect in high-risk patients already treated with BCG, or whether its impact is more pronounced in lower-risk groups. This represents a key area for future risk-adjusted research.

#### 4.3.10. Timing of BOO Surgery (TURBT + HoLEP)

The optimal timing of BOO resolution in NMIBC remains insufficiently defined. In clinical practice, most urologists prioritize complete and accurate TURBT, followed by risk-adapted intravesical therapy, and schedule BOO surgery (TURP or HoLEP) either between induction and maintenance cycles or after the completion of induction, once pathology and risk status are fully established. Small series have reported the feasibility of simultaneous TURBT and BOO surgery, but concerns persist regarding perioperative bleeding, tumor seeding, and impaired healing in the setting of large resection areas. Current evidence does not allow us to recommend routine same-session HoLEP and TURBT; instead, a staged approach individualized to tumor risk, symptom burden, and anesthesia risk seems prudent until more robust data become available.

#### 4.3.11. Clinical Implications

From a practical standpoint, clinicians should routinely assess BOO in NMIBC patients using objective measures such as PVR, uroflowmetry, and symptom indices. Addressing clinically significant BOO has clear functional benefits: it reduces urinary retention, improves LUTS, and enhances patient tolerance of intravesical therapy, which is often poorly tolerated in obstructed bladders. Importantly, while decompression may indirectly support intravesical treatment performance, there is currently no evidence that any specific BOO treatment—HoLEP, TURP, or pharmacotherapy—confers superior oncological benefit. Therefore, the choice of intervention should be guided by established functional outcomes, prostate size, surgeon expertise, and patient comorbidity. At present, HoLEP may offer greater durability and more complete anatomic relief of BOO, but it should not be recommended over TURP or medical therapy on the basis of presumed cancer control. Until prospective risk-adjusted studies are available, BOO management in NMIBC should focus on optimizing quality of life and improving intravesical therapy tolerance rather than altering oncologic prognosis.

#### 4.3.12. Limitations and Future Directions

Despite biologically plausible mechanisms and encouraging early clinical data, several methodological and interpretative limitations temper the generalizability of current findings. Almost all studies are retrospective or observational, thresholds for PVR and definitions of BOO vary, oncologic endpoints and follow-up intervals are not standardized, and study populations are heterogeneous with limited risk-stratified analyses. Furthermore, there are no human studies correlating PVR with inflammatory biomarkers or intravesical drug pharmacokinetics in NMIBC, reinforcing that much of the pathophysiologic discussion remains inferential. In particular, confounding by indication—whereby younger, fitter patients with more favorable disease or anatomy preferentially receive HoLEP—may fully account for the lower recurrence proportions reported in retrospective series, underscoring that these findings should not be interpreted as evidence of a causal protective effect. Collectively, these limitations mean that all reported associations between BOO, HoLEP, and NMIBC outcomes should be regarded as preliminary and hypothesis-generating, rather than as evidence of a proven causal relationship. The comparative study by Garg et al. reporting lower recurrence following HoLEP should be interpreted cautiously. Patients selected for HoLEP had more favorable baseline characteristics, including younger age, lower comorbidity, and anatomy more suitable for complete enucleation—features that conferred a better prognosis independent of surgical modality. Because these differences were neither randomized nor fully adjusted for tumor biology or intravesical therapy intensity, the reported associations should be attributed to selection bias until proven otherwise. The study therefore offers a hypothesis-generating observation rather than evidence of a causal oncologic effect.

#### 4.3.13. Future Research Priorities

To advance the field and bridge current knowledge gaps, several strategic directions should be prioritized:Prospective Randomized Controlled Trials (RCTs):

Rigorous RCTs comparing NMIBC patients with BOO undergoing HoLEP vs. those receiving TURP, medical therapy, or no BOO treatment are essential. These trials should assess

○Recurrence-free survival;○Progression rates;○Intravesical treatment efficacy;○Bladder function over time.

Ideally, patients should be stratified based on tumor risk groups and BOO severity to enable personalized insights.

2.Patient Stratification by Functional and Anatomical Markers:

Future studies should incorporate detailed stratification using

○PVR volumes (e.g., <50 mL, 50–100 mL, >100 mL);○LUTS severity (e.g., IPSS or other validated scales);○Bladder wall morphology (e.g., presence of trabeculation, diverticula) assessed via imaging or cystoscopy.

This stratification will allow a granular understanding of which patient subgroups derive the greatest oncological benefit from BOO treatment.

3.Longitudinal Evaluation of Functional and Quality-of-Life Outcomes:

While recurrence and progression are primary endpoints, it is equally important to evaluate

○Durability of PVR reduction and bladder compliance;○Postoperative urinary continence and sexual function;○Patient-reported outcomes and treatment satisfaction.

These data will be critical in informing shared decision-making in older patients or those with competing health risks.

4.Incorporation of Biomarkers and Mechanistic Studies:

Integrating urinary cytokine assays, urothelial gene expression profiles, and bladder tissue histology into future studies may provide insight into the inflammatory and immunological pathways affected by BOO and its resolution. These markers could serve as predictors of recurrence risk or response to BCG therapy.

5.Cost-Effectiveness and Health Economics Analyses:

Given the higher upfront resource demands of HoLEP (equipment, training, operative time), future investigations should assess the cost-effectiveness of HoLEP compared to TURP or conservative management in NMIBC patients. This is particularly relevant in healthcare systems with limited access to laser technology.

6.Multicenter Collaborations and Registry Data:

Leveraging national and international registries of NMIBC patients could facilitate large-scale, real-world evidence generation. Pooling data from high-volume HoLEP centers may enable robust propensity-matched comparisons and facilitate long-term surveillance of oncologic outcomes in a pragmatic setting.

## 5. Conclusions

Chronic urinary retention, most often resulting from bladder outlet obstruction (BOO), is an underappreciated but potentially modifiable factor in the management of non-muscle-invasive bladder cancer (NMIBC). By prolonging urothelial exposure to carcinogens, exacerbating inflammation, and impairing intravesical therapy delivery, urinary retention may contribute to higher recurrence rates and reduced treatment tolerability.

Holmium Laser Enucleation of the Prostate (HoLEP) provides durable, size-independent relief of BOO, reliably reduces PVR, and improves LUTS and quality of life in men with benign prostatic hyperplasia. Early, retrospective data suggest an association between HoLEP-mediated BOO resolution and lower NMIBC recurrence in selected patients; however, these observations are subject to confounding and do not establish causality.

At present, HoLEP should therefore be regarded as a promising functional adjunct for carefully selected NMIBC patients with significant BOO, rather than as a proven oncologic intervention. Future prospective, multicenter, risk-adjusted studies—including standardized PVR thresholds, clearly defined risk groups, and integrated functional and oncologic endpoints—are essential to determine whether BOO correction can translate into durable improvements in NMIBC recurrence, progression, and survival. Accordingly, all associations between BOO correction (including HoLEP) and improved NMIBC outcomes should be interpreted as preliminary and hypothesis-generating; current data do not justify changes to guideline-recommended oncologic management outside of clinical trials or carefully individualized decision-making. Accordingly, our conclusions are intentionally conservative: current evidence suggests a possible link between BOO correction and improved NMIBC control, but definitive evaluation of causal impact will require prospective, multicenter, risk-adjusted studies incorporating standardized PVR thresholds, risk groups, and integrated functional and oncologic endpoints.

## Figures and Tables

**Table 1 cancers-17-03864-t001:** Key studies informing the synthesis.

Key Studies Informing the Synthesis											
Study (Year, Ref.)	Design	N	Population	BOO/Retention Metric	Threshold(s)	Intervention/Comparator	Primary Outcomes	Key Findings	Effect Estimates (OR/HR/RR)	Follow-Up	Key Limitations/Risk of Bias
Sazuka et al., 2020 [21]	Retrospective cohort	356	NMIBC after TURBT	PVR (US)	>100 vs. ≤100 mL	Standard care	Intravesical recurrence	High PVR independently ↑ recurrence	**Adjusted HR ≈ 1.9** (reported in study)	36 mo	Retrospective; heterogeneous IVT regimens; no standardized PVR schedule; no risk-stratified survival; residual confounding likely.
Lunney et al., 2019 [3]	Retrospective cohort	164	NMIBC post-TURBT	LUTS (IPSS)	Mod/Severe vs. Mild	Standard care	Recurrence	LUTS severity independently predicted recurrence	**OR 19.1** (moderate/severe vs. mild); **OR 1.26 per IPSS point**	24 mo	Subjective BOO proxy; no objective PVR; variable recurrence assessment; incomplete adjustment for cofounders (tumor burden, smoking).
Di Gianfrancesco et al., 2023 [4]	Multicenter cohort	312	NMIBC on BCG or MMC	PVR	>80 vs. ≤80 mL	BCG/MMC	RFS, PFS	High PVR → shorter RFS & PFS	**Elevated PVR = independent predictor of non–tumor-free status** (effect size not numerically provided)	24–36 mo	Multicenter heterogeneity; unvalidated PVR cutoff; selection bias (who received PVR evaluation); no risk-adjusted HR reported.
Yentur et al., 2025 [2]	Retrospective cohort	228	Low-risk Ta NMIBC	BOO ± PVR	Center-defined	Standard care	Recurrence	BOO/retention associated with higher recurrence	No OR/HR reported	24 mo	Center-specific BOO definitions; heterogeneous IVT; no effect estimates; absence of multivariable or subgroup analyses.
Can et al., 2025 [5]	Retrospective cohort	301	NMIBC	BOO (urodynamic/clinical)	NA	Standard	Recurrence, progression; bladder remodeling	BOO not predictor of progression; bladder remodeling correlated w/higher grade	**Trabeculation** → **OR 4.62 for high-grade/CIS**	36 mo	Remodeling not quantified uniformly; confounding by duration of obstruction; limited progression follow-up; observational design.
Cicione et al., 2018 [22]	Multicenter cohort	437	NMIBC	Detrusor Wall Thickness	>2.5 mm	Standard	Recurrence, progression	Increased DWT predicts worse outcomes	**OR 4.9 for recurrence; OR 2.21 for progression**	24–60 mo	No direct BOO/PVR measures; DWT etiology unclear; heterogeneous treatment; absence of confounder adjustment (risk group, IVT).
Porreca et al., 2025 [23]	Retrospective comparative	300	NMIBC + BOO	PVR/LUTS	NA	HoLEP vs. no surgery	Recurrence, progression	HoLEP group showed lower recurrence and progression	OR 0.65 for recurrence	36 mo	Multicentric heterogeneity, relatively short follow-up period (long term recurrence and progression rates require further evaluation)
Dai et al., 2016 [24] (Medicine)	Meta-analysis	9 studies	General BPH/BOO	Clinical BOO/BPH	NA	BPH vs. non-BPH	Bladder cancer incidence	BPH associated with ↑ bladder cancer risk	**Case–control OR 2.50; Cohort RR 1.58**	NA	Non-NMIBC specific; observational source studies; no adjustment for retention severity; reverse causality possible.
Sun et al., 2014 [9]	RCT	205	BOO/BPH	PVR, Qmax, IPSS	NA	HoLEP vs. TURP	Functional outcomes	HoLEP superior for decompression, retreatment	Not oncologic	60 mo	BPH population only; no cancer outcomes; external validity to NMIBC indirect.
Gilling et al., 2017 [8]	Longitudinal registry	949	BOO/BPH	PVR, Qmax, IPSS	NA	HoLEP	Symptom, PVR durability	Durable PVR ↓ (<50 mL)	Not oncologic	Up to 120 mo	Registry; attrition bias; no oncologic data; extrapolation to NMIBC functional only.
Chen et al., 2022 [25]	Systematic review	>2600	BOO/BPH	PVR/Qmax/IPSS	NA	HoLEP vs. TURP	Efficacy & safety	HoLEP ≥ TURP; lower retreatment	Not oncologic	Varies	Functional-only; pooled BPH cohorts; heterogeneity; no NMIBC-relevant endpoints.
Lee et al., 2024 [26]	Prospective registry	1028	BOO/BPH	PVR, Qmax, IPSS	NA	HoLEP	Functional & safety	Consistent PVR ↓, Qmax ↑	Not oncologic	12–24 mo	No cancer patients; registry selection; short follow-up relative to cancer recurrence timelines.
Porreca et al., 2021 [6]	Single center	577	BOO/BPH	PVR	NA	HoLEP (technique)	Functional & peri-op	Symptom/PVR improvement	Not oncologic	12 mo	Technique report; no oncologic outcomes; single-center; selection bias.

**Table 2 cancers-17-03864-t002:** Functional Impact of HoLEP on Bladder Emptying: Detailed PVR Outcomes From Large BOO/BPH Cohorts Relevant to NMIBC Management.

Study (Year)	Design/N	Baseline PVR (mL)	Post-HoLEP PVR (mL) & Timepoint	Effect Size/Comparative Data	Relevance to NMIBC Context
Lee et al. [26]	Prospective multicenter registry; *n* = 3000	51.0 (IQR 20–109)	3 mo: 6.0 (IQR 0–31.5); 6 mo: 2.0 (IQR 0–27.0)	Absolute median reduction ≈ 45–50 mL	Demonstrates reliable normalization of PVR after anatomically complete enucleation; foundational evidence for BOO-correction–based hypothesis in NMIBC.
Morozov et al. [29]	Large single-center cohort; *n* = 509	70.8	17.2 at 6 months	Absolute reduction ≈ 53.6 mL	Reinforces reproducibility of PVR improvement across centers; supports the biological rationale that resolving retention reduces chronic urothelial irritation.
Chen et al. [25]	Systematic review & meta-analysis of 8 RCTs (HoLEP vs. TURP)	— (varies per RCT)	HoLEP showed consistently lower PVR vs. TURP at 6 mo (MD −9.78 mL) and 12 mo (MD −9.93 mL)	HoLEP superior to TURP in sustained PVR reduction	ND: Supports the concept that degree of BOO relief differs by technique, which may influence intravesical therapy performance in NMIBC.

PVR = postvoid residual; BOO = bladder outlet obstruction; BPH = benign prostatic hyperplasia; MD = mean difference. Note: These studies involve BPH/BOO cohorts without known bladder cancer. They are included to document the reproducible magnitude of PVR reduction achievable with HoLEP across diverse populations and study designs. Because elevated PVR and chronic urinary retention are emerging prognostic modifiers in NMIBC, these functional outcomes provide essential mechanistic context for the hypothesized effect of BOO correction on NMIBC recurrence and treatment efficacy.

**Table 3 cancers-17-03864-t003:** Summary of Key Domains: Established Evidence vs. Emerging/Hypothesis-Generating Evidence.

Domain	Established Evidence	Emerging/Hypothesis-Generating Evidence
Epidemiology of NMIBC	NMIBC represents 70–75% of incident bladder cancers. High recurrence rates (50–70%); progression in 10–20% of high-risk cases. Major economic burden driven by lifelong surveillance and repeated interventions.	—
Pathophysiological impact of chronic urinary retention (CUR)	CUR causes prolonged urothelial exposure to urinary solutes. BOO leads to elevated PVR, bladder wall remodeling, and reduced compliance.	CUR-related inflammation may foster a microenvironment permissive for recurrence and intravesical therapy resistance. Drug-distribution alterations due to large PVR may reduce BCG/chemotherapy efficacy.
Clinical evidence linking BOO/PVR to NMIBC outcomes	Elevated PVR (>80–100 mL) and moderate–severe LUTS independently predict recurrence after TURBT. High PVR associated with reduced BCG/mitomycin response.	BOO-related bladder remodeling (trabeculation, increased DWT) may correlate with higher-grade tumors or increased recurrence risk. BOO as a potential modifier of NMIBC biology remains to be validated prospectively.
Inflammatory & immunological effects	Chronic LUT inflammation increases urothelial cytokines (TNF-α, IL-6, IL-8). Intravesical BCG induces sustained cytokine production.	Inflammation-induced barrier dysfunction may impair immune-mediated tumor cytotoxicity. TNF-α–driven transcriptional changes may influence BCG response (mechanistic plausibility only).
Functional parameters as prognostic indicators	LUTS severity and PVR are reproducible markers of voiding dysfunction.	Emerging data suggest PVR, compliance, and remodeling markers may serve as prognostic variables in NMIBC recurrence pathways.
HoLEP functional outcomes	HoLEP provides durable, size-independent relief of BOO with major reductions in PVR and sustained improvements in Qmax/IPSS. Lower retreatment rates vs. TURP.	Degree of anatomical decompression may influence intravesical therapy distribution and tolerability.
Oncologic impact of HoLEP in NMIBC	—	Retrospective studies report lower recurrence after HoLEP vs. TURP/no intervention, but findings are confounded (younger, fitter surgical candidates). No evidence of causality; current data are hypothesis-generating.
HoLEP vs. TURP	TURP effective but less durable, particularly in large prostates. HoLEP associated with fewer complications, lower retreatment rates, and more complete enucleation.	Preliminary data suggest HoLEP may reduce recurrence more effectively than TURP, but confounding by indication is substantial.
Quality of life, adherence, intravesical therapy tolerance	Intravesical therapy frequently exacerbates LUTS, reducing adherence.	HoLEP may mitigate therapy-related LUTS and improve intravesical treatment tolerance by reducing PVR—but prospective validation is needed.
Limitations of current evidence	Available studies are largely retrospective. Heterogeneous follow-up durations and inconsistent PVR thresholds.	Lack of integrated functional + oncologic endpoints. No biomarker-based validation of BOO-related mechanisms in NMIBC.
Future research priorities	—	Prospective RCTs comparing HoLEP vs. TURP/medical management in NMIBC with BOO. Stratification by PVR, LUTS severity, DWT/trabeculation. Biomarker studies (cytokines, immune signatures). Cost-effectiveness analyses and multicenter longitudinal registries.
Future Research Directions	Prospective RCTs comparing HoLEP vs. TURP/medical therapy in NMIBC patients with BOO. Stratification by PVR, LUTS severity, bladder morphology. Longitudinal evaluation of functional + oncologic outcomes. Integration of urinary biomarkers (cytokines, immune markers). Cost-effectiveness analyses and multicenter registry studies.	—
Conclusion	CUR is a modifiable risk factor in NMIBC recurrence and therapy resistance. HoLEP provides durable BOO resolution, functional recovery, and potential oncologic benefit. Should be considered as part of integrated NMIBC management pending stronger prospective evidence.	—

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
