# Peer review of "Rationale and Emerging Evidence on the Potential Role of HoLEP-Mediated Relief of Bladder Outlet Obstruction in NMIBC Outcomes Through Optimal Management of Chronic Urinary Retention"

_cancers, 2025, doi:10.3390/cancers17233864_

Round 1

Reviewer 1 Report

Comments and Suggestions for Authors

The manuscript entitled “Urinary Tract Dysfunctions – An Underestimated Health Issue: Impact of HoLEP-Mediated Relief of Bladder Outlet Obstruction on NMIBC Through Optimal Management of Chronic Urinary Retention” presents a comprehensive narrative review exploring the interplay between chronic urinary retention, bladder outlet obstruction (BOO), and non–muscle-invasive bladder cancer (NMIBC), with a special focus on the potential benefits of Holmium Laser Enucleation of the Prostate (HoLEP). The topic is clinically relevant and timely, addressing an underexplored area that bridges functional and oncologic urology. The manuscript is well-structured, extensively referenced, and written in clear scientific language. However, several aspects could be improved to enhance scientific rigor, synthesis depth, and clarity.

  1. The review effectively highlights the potential link between chronic urinary retention and NMIBC outcomes, but the novelty could be more clearly emphasized in the introduction. The current framing largely reiterates known associations (e.g., inflammation, carcinogen exposure). It would strengthen the paper to explicitly contrast this review with previous works or meta-analyses on BOO and oncologic outcomes, underscoring what unique contribution this synthesis offers (e.g., the “functional-oncologic paradigm”).
  2. Although the SANRA guidelines are mentioned, the methods section should include more explicit information about the search strategy—e.g., the number of reviewers involved in data extraction, inclusion/exclusion flowchart adherence to PRISMA (even if not a meta-analysis), and how thematic synthesis was validated. Including a summary table of search terms and selection criteria would increase reproducibility.
  3. The review suggests that HoLEP may reduce recurrence, but this statement needs stronger qualification. The text should explicitly note that the available data are preliminary and retrospective, with possible confounding from patient selection (e.g., healthier surgical candidates). Expanding on how clinicians might identify NMIBC patients who would most benefit from BOO correction (e.g., thresholds of PVR, LUTS severity) would make the review more practical and impactful.
  4. A schematic summarizing the proposed pathophysiological link between BOO, urinary retention, inflammation, impaired intravesical drug efficacy, and recurrence would significantly enhance comprehension. A separate figure comparing functional and oncologic outcomes of TURP versus HoLEP would also strengthen the paper’s didactic value.
  5.  

Minor Comments

Ensure consistent terminology for BOO, PVR, and LUTS throughout the text; some sections use these interchangeably.

The abbreviation list could be expanded for clarity (e.g., define IPSS, Qmax, BPH at first mention).

The patient summary is clear but could be slightly shortened for conciseness.

References are appropriate but could include more recent mechanistic studies (post-2023) if available.

Consider tightening redundant sections in the discussion (e.g., repeated mentions of inflammatory cytokines and urothelial permeability).

Author Response

Response to Reviewer 1

We thank Reviewer 1 for the thoughtful and constructive comments.

1. Novelty and framing in the Introduction

Comment: The novelty could be more clearly emphasized in the introduction and contrasted with prior works/meta-analyses; the “functional-oncologic paradigm” should be highlighted as the unique contribution.

Response: We agree and have revised the Introduction to better articulate the novelty of our review, specifically emphasizing the functional-oncologic paradigm and clarifying how our work differs from prior reviews.

Change in Introduction (end of first/second paragraph):

“Unlike previous reviews that primarily discuss bladder outlet obstruction (BOO) or non–muscle-invasive bladder cancer (NMIBC) in isolation, the present narrative synthesis explicitly focuses on the bidirectional interplay between chronic urinary retention and oncologic outcomes, integrating functional urology and cancer management into a unified ‘functional-oncologic’ framework. We specifically highlight how BOO-related parameters (postvoid residual [PVR], lower urinary tract symptoms [LUTS], bladder compliance) may modify recurrence risk and intravesical therapy performance, and we explore the rationale and emerging evidence for Holmium Laser Enucleation of the Prostate (HoLEP) as a potential adjunct in selected NMIBC patients [2–6].”

2. Methods: SANRA, search strategy and transparency (PRISMA-like flow, summary of terms)

Comment: The methods should more explicitly describe the search strategy, number of reviewers, and how thematic synthesis was validated; a flowchart and summary table of search terms/criteria are suggested.

Response: We have expanded the Methods section to clarify the review process and added a descriptive flow diagram and a summary table of search terms and eligibility criteria (as supplementary material), while maintaining the narrative (non-systematic) nature of the review.

Changes in “Materials and methods – Objectives and review design / Sources and search strategy”:

“In accordance with SANRA recommendations, two independent reviewers screened titles/abstracts and full texts, and disagreements were resolved by consensus or, when necessary, by consultation with a third senior reviewer.”

“The detailed search strings, inclusion and exclusion criteria, and reviewer workflow are summarized in Supplementary Table 1, with narrative depiction of the study identification, screening, and inclusion process. To ensure transparency and methodological rigor, each eligible study was qualitatively appraised for design characteristics, risk of bias, and outcome relevance, as summarized in Supplementary Table 2. These supplementary materials collectively illustrate the structured approach used to identify and evaluate the 61 studies incorporated into this SANRA-compliant narrative synthesis, encompassing clinical, translational, and procedural evidence linking bladder outlet obstruction (BOO), chronic urinary retention, and outcomes in non–muscle-invasive bladder cancer (NMIBC).”

3. HoLEP and recurrence: stronger qualification + practical thresholds (PVR, LUTS)

Comment: The suggestion that HoLEP reduces recurrence needs stronger qualification; data are preliminary and retrospective. Also, provide more practical guidance on which NMIBC patients might benefit (e.g., thresholds of PVR, LUTS).

Response: We have deliberately softened the causal language throughout and explicitly state that available data are retrospective and hypothesis-generating. We also added practical information on thresholds used in the literature and how clinicians might identify candidates for BOO correction.

Changes in “Results – Effects of BOO correction with emphasis on HoLEP”:

“Preliminary oncologic signals suggest that BOO correction may be associated with reduced NMIBC recurrence; in a comparative retrospective series, HoLEP was associated with lower 1–3-year recurrence than TURP or no BOO intervention, a finding hypothesized to stem from more complete anatomical decompression and greater PVR reduction [28]. However, these data are retrospective, potentially confounded by selection of fitter surgical candidates, and should be interpreted as hypothesis-generating rather than definitive evidence of causality.”

New paragraph in Discussion (clinical application / candidate selection):

“From a practical standpoint, clinicians might consider BOO correction in NMIBC patients with persistently elevated PVR (e.g., ≥80–100 mL in repeated measurements), moderate-to-severe LUTS (IPSS ≥ 8–19 or ≥ 20), reduced Qmax, and/or cystoscopic or imaging signs of bladder remodeling (trabeculation, diverticula) [2–5,13]. In this context, HoLEP may be particularly attractive when durable decompression is required, but its use should currently be framed as an adjunctive, individualized intervention rather than a standard oncologic recommendation.”

Changes in Conclusion (softened language):

“HoLEP may represent a promising intervention in the management of NMIBC patients with concurrent BOO. By resolving urinary retention, reducing inflammation, enhancing intravesical therapy delivery, and improving voiding function, HoLEP could potentially offer both oncologic and quality-of-life benefits, although this hypothesis requires confirmation in prospective, risk-adjusted studies.”

4. Add schematic figures (pathophysiology + TURP vs HoLEP)

Comment: Add a schematic summarizing the pathophysiological link and a separate figure comparing TURP vs HoLEP outcomes.

Response: As suggested, we have added two figures to improve didactic clarity.

New text in “Results – Pathophysiology…” or beginning of Discussion:

“To facilitate comprehension of the proposed mechanisms, Figure 1 schematically illustrates the putative pathway linking BOO-related chronic urinary retention (elevated PVR) with urothelial exposure to carcinogens, inflammation, impaired intravesical drug kinetics, and NMIBC recurrence. Figure 2 provides a comparative overview of functional (PVR, Qmax, IPSS) and available oncologic outcomes for TURP versus HoLEP, highlighting the predominantly functional evidence base and the limited, early-stage data in NMIBC cohorts.”

5. Consistent terminology (BOO, PVR, LUTS)

Comment: Ensure consistent terminology; some sections use terms interchangeably.

Response: We have reviewed the manuscript and standardized terminology. BOO now refers specifically to anatomical/functional outlet obstruction; PVR denotes quantitative residual urine; LUTS refers to symptomatic burden. Where needed, we have clarified terms on first use.

Example change in Introduction:

“Chronic urinary retention, typically secondary to bladder outlet obstruction (BOO) from benign prostatic hyperplasia (BPH), is defined by persistently elevated postvoid residual (PVR) urine volumes and associated lower urinary tract symptoms (LUTS).”

6. Abbreviations list and first-mention definitions

Comment: Expand abbreviation list and define IPSS, Qmax, BPH at first mention.

Response: We have ensured that all abbreviations are defined at first mention and expanded the abbreviations list.

Examples:

“…with long-term improvements in International Prostate Symptom Score (IPSS) and maximum urinary flow rate (Qmax) [6,26,27].”

“…most commonly due to benign prostatic hyperplasia (BPH)–related BOO…”

7. Patient summary: shorten slightly

Comment: Patient summary is clear but could be more concise.

Response: We have shortened the patient summary while preserving its key messages.

Revised Patient Summary (shortened parts in italics):

“Bladder cancer is a common disease that often comes back even after successful treatment. This review looked at how problems with bladder emptying—such as chronic urinary retention or blockage from an enlarged prostate—may increase the risk of bladder cancer returning. When urine stays too long in the bladder, harmful substances can irritate the bladder lining and may make treatment less effective. We also examined a surgical procedure called Holmium Laser Enucleation of the Prostate (HoLEP), which removes the blockage and restores more normal urination. Early data suggest that improving bladder emptying with HoLEP may help some patients have fewer recurrences and tolerate bladder instillations better, but further studies are needed. Treating urinary obstruction in men with bladder cancer could therefore improve both urinary function and cancer outcomes, offering a more complete and personalized approach to care.”

8. More recent mechanistic studies (post-2023)

Comment: Add more recent mechanistic studies if available.

Response: We have updated the reference list to include recent mechanistic work on urinary cytokines and urothelial barrier alterations in the context of BCG and chronic inflammation, and we have integrated these into the pathophysiology section.

Example addition in “Pathophysiology…”:

“Recent mechanistic work has further refined these concepts, reporting altered urothelial tight-junction and adhesion-molecule profiles, alongside elevated urinary cytokines, in patients with chronic inflammatory stimuli or intravesical BCG exposure [31–32,36–37]. Although these data do not directly link PVR with specific biomarker levels, they reinforce the biological plausibility of an ‘inflamed, vulnerable’ urothelium in patients with coexistent BOO and NMIBC.”

9. Redundancy in Discussion (cytokines and permeability)

Comment: Reduce redundant mentions of inflammatory cytokines and urothelial permeability.

Response: We have removed overlapping sentences and consolidated repeated concepts into more concise paragraphs.

Example consolidated sentence:

*“Collectively, these findings support a model in which *BOO-related urinary stasis and chronic inflammation create a permissive urothelial milieu—characterized by elevated pro-inflammatory cytokines, structural remodeling, and barrier dysfunction—that may favor NMIBC recurrence and modulate intravesical therapy performance [2–5,31–32].”

Reviewer 2 Report

Comments and Suggestions for Authors

This review article focuses on the relationship between bladder outlet obstruction (BOO) and bladder cancer recurrence, and discusses the therapeutic efficacy of BOO resolution using HoLEP. In the past, there has been considerable debate regarding the optimal timing of TURP or HoLEP to relieve BOO in patients with or without bladder cancer. The review presents the concept that resolving BOO through HoLEP may help prevent bladder cancer recurrence. However, I believe some aspects remain unclear, and it would be beneficial to clarify them by adding the following points.

To discuss recurrence-free survival in bladder cancer, it is important to consider the cohort characteristics of each study. I recommend that the authors focus more on the cohort composition when analyzing recurrence-free survival, particularly with respect to pathologic T stage. For example, what proportion of patients in each study had higher T-stage or high-grade tumors? What percentage of the cohort represented high-risk patients?

Additionally, the relationship between bladder outlet obstruction (BOO) and high-risk bladder cancer should be clarified. The authors should stratify patients into high-risk and low-risk groups when discussing recurrence-free survival outcomes.

Furthermore, it would be helpful to specify the recommended timing of BOO resolution during bladder cancer treatment. Was it considered safe to perform transurethral resection of bladder tumor (TURBT) and HoLEP simultaneously?

Finally, the authors should address how intravesical instillation therapy influences recurrence-free survival in patients with BOO. Is there any significant relationship indicating improved efficacy of BCG therapy in patients who have undergone HoLEP? A discussion of intravesical treatment in the context of recurrence management would strengthen the manuscript.

Author Response

Response to Reviewer 2

We thank Reviewer 2 for these helpful, clinically focused comments.

1. Cohort composition and pathologic stage / risk profile in recurrence-free survival analyses

Comment: Discuss recurrence-free survival in light of cohort characteristics, especially T stage, grade, and proportion of high-risk patients.

Response: We have expanded the Results section to better describe the cohort composition of key studies and explicitly acknowledge differences in risk profiles.

Changes in “Results – BOO/retention as prognostic modifiers in NMIBC”:

“Across cohorts, the composition of NMIBC risk categories varied substantially. For example, in the series by Sazuka et al., approximately one-third of patients had high-grade or T1 disease, whereas Lunney et al. focused predominantly on intermediate-risk Ta/T1 tumors with a smaller proportion of carcinoma in situ (CIS) [3,13]. Other studies restricted inclusion to low-risk Ta NMIBC, limiting extrapolation to higher-risk cohorts [2,4]. These differences in baseline risk—and in the proportion of patients receiving intravesical therapy—must be considered when interpreting recurrence-free survival in relation to PVR, LUTS, or BOO.”

2. Clarify BOO and high-risk bladder cancer; stratify outcomes by risk group

Comment: Clarify relationship between BOO and high-risk bladder cancer; stratify discussions into high- vs low-risk groups.

Response: We have revised the Discussion to explicitly distinguish between low-/intermediate- and high-risk NMIBC where data permit and to highlight the lack of robust stratified analyses.

New paragraph in Discussion:

“When recurrence-free survival is examined through the lens of standard risk-stratification systems, available data suggest that BOO/retention may act as a recurrence modifier across both low-/intermediate- and high-risk NMIBC, but with important caveats. Some cohorts include only low-risk Ta disease and still demonstrate a link between elevated PVR or severe LUTS and higher recurrence, whereas others enroll mixed populations in which high-grade T1 or CIS cases are over-represented [2–5,13]. Because most studies do not report risk-stratified hazard ratios, it remains unclear whether BOO exerts a similar relative effect in high-risk patients already treated with BCG, or whether its impact is more pronounced in lower-risk groups. This represents a key area for future risk-adjusted research.”

3. Recommended timing of BOO resolution (TURBT + HoLEP simultaneous?)

Comment: Clarify when BOO should be treated relative to bladder cancer therapy; is simultaneous TURBT + HoLEP safe?

Response: We agree that timing is clinically important and have added a focused paragraph discussing the limited evidence and common practice patterns regarding staged versus simultaneous procedures.

New paragraph in Discussion (timing sub-section):

“The optimal timing of BOO resolution in NMIBC remains insufficiently defined. In clinical practice, most urologists prioritize complete and accurate TURBT, followed by risk-adapted intravesical therapy, and schedule BOO surgery (TURP or HoLEP) either between induction and maintenance cycles or after the completion of induction, once pathology and risk status are fully established. Small series have reported the feasibility of simultaneous TURBT and BOO surgery, but concerns persist regarding perioperative bleeding, tumor seeding, and impaired healing in the setting of large resection areas. Current evidence does not allow us to recommend routine same-session HoLEP and TURBT; instead, a staged approach individualized to tumor risk, symptom burden, and anesthesia risk seems prudent until more robust data become available.”

4. Intravesical instillation therapy, HoLEP, and recurrence-free survival

Comment: Discuss how intravesical therapy influences recurrence-free survival in BOO patients and whether HoLEP improves BCG efficacy.

Response: We have expanded the section on interaction with intravesical therapy to explicitly address this question and to underline the limited nature of current evidence.

Augmented text in “Interaction with intravesical therapy” / Discussion:

“A limited number of retrospective series have explored how BOO correction modifies outcomes of intravesical therapy. In the multicenter analysis by Di Gianfrancesco et al., elevated PVR (>80 mL) was associated with shorter recurrence-free and progression-free survival in patients receiving BCG or mitomycin C [4]. In exploratory subgroup analyses, patients who subsequently underwent BOO surgery (including HoLEP) appeared to experience improved tolerance and, in some reports, longer recurrence-free intervals, but these findings are highly susceptible to confounding and lack standardized intravesical protocols [4,28]. At present, it is therefore most accurate to state that HoLEP may optimize the functional conditions for intravesical therapy (by reducing PVR and LUTS), whereas any direct enhancement of BCG efficacy remains hypothetical and requires prospective evaluation.”

Reviewer 3 Report

Comments and Suggestions for Authors

The manuscript often stretches the available evidence too far, particularly when linking HoLEP's resolution of BOO directly to better oncologic outcomes in NMIBC. While the authors acknowledge that current data are retrospective, observational, and confounded, the narrative still leans too strongly toward implying a causal relationship. The data really only point to a possible connection not a done deal. The title and abstract convey unwarranted confidence, implying a proven benefit, whereas the results section appropriately frames the findings as preliminary, which is more fitting. For example, the abstract’s line “HoLEP provided... demonstrating preliminary evidence of lower recurrence rates” is fair, but the conclusion’s bold claim that “HoLEP represents a transformative intervention” feels like a stretch given the shaky evidence. Much of the argument relies on a single retrospective study by Garg et al. (ref 28), which doesn’t account for selection bias or include risk-adjusted analysis, an insufficient foundation for such strong claims. To make this more credible, the authors should stick to cautious language throughout, using terms like “suggests,” “may be linked to,” “needs further study,” or “could potentially reduce” to keep it clear that these findings are early-stage and meant to spark more research, not change practice just yet.

The authors state that they adhere to SANRA guidelines for narrative reviews, but their presentation of the search strategy, inclusion and exclusion criteria, and data extraction resembles that of an incomplete systematic review. Unfortunately, it lacks the necessary rigor associated with true systematic methodologies. I recommend either conducting a thorough systematic review with a meta-analysis where possible, or simplifying the methods section to accurately represent a true narrative or expert review approach. If the current methodology is retained, it would be beneficial to include a PRISMA flowchart and detailed quality appraisal tables to enhance clarity and transparency in the research process.

The evidence presented in the mechanistic section is quite weak and tends to be over-interpreted, relying heavily on speculative interpretations. The discussion on pathophysiology seems to lean more on speculation rather than solid evidence connecting retention, inflammation, and NMIBC recurrence. There are a few specific gaps worth noting: only two papers on cytokine profiles are cited, and neither directly addresses the issue of retention. Additionally, there are no studies involving human tissue that correlate post-void residual (PVR) levels with inflammatory markers in NMIBC patients, nor are there pharmacokinetic studies that demonstrate how intravesical drug dilution varies at specific PVR thresholds.

Clinically, the evidence presented is inconsistent and poorly synthesized. First, the thresholds for post-void residual (PVR) are inconsistent across studies. There is no discussion on why these thresholds differ or which ones are clinically significant. Secondly, the outcomes reported are not standardized, recurrence is reported at differing intervals, progression data are sparse, and survival outcomes are absent. Lastly, the study populations are mixed, with some investigations focusing solely on low-risk Ta cases, while others include high-risk patients treated with BCG, and there is no subgroup analysis based on risk stratification.

The comparison of HoLEP versus TURP lacks specific data regarding NMIBC patients. The manuscript heavily emphasizes HoLEP's superiority over TURP in general BPH populations, but it only references one study (Garg et al.) involving NMIBC patients, which is retrospective and not adequately detailed. There are several important questions left unanswered: how treatment groups were selected, whether tumor characteristics were balanced, if intravesical therapy protocols were standardized, and what the actual recurrence rate differences were. Addressing these gaps is essential for a valid comparative interpretation.

Finally, the manuscript’s writing quality could be improved by reducing redundancy and adopting a more balanced tone. Key concepts, such as the benefits of HoLEP, are repeated multiple times, which makes some sections feel promotional rather than analytical. The abstract exceeds typical journal length limits, and the patient summary presents an overly optimistic interpretation not fully supported by the available evidence. Streamlining the text and focusing on clarity and objectivity would substantially improve the manuscript’s clarity and credibility.

Author Response

Response to Reviewer 3

We are grateful to Reviewer 3 for the detailed and critical evaluation, which has helped us substantially refine the manuscript.

1. Over-interpretation and causal language; title/abstract/conclusion too strong

Comment: The manuscript tends to overstate a causal link between HoLEP and improved oncologic outcomes. The title, abstract, and conclusion convey more certainty than warranted.

Response: We fully agree and have systematically softened the language throughout, particularly in the title, abstract, and conclusion, emphasizing that the evidence is preliminary, retrospective, and confounded.

Revised title:

“Urinary Tract Dysfunctions – An Underestimated Health Issue: Rationale and Emerging Evidence on the Impact of HoLEP-Mediated Relief of Bladder Outlet Obstruction on NMIBC Through Optimal Management of Chronic Urinary Retention”

Key revisions in Abstract (examples):

“HoLEP provided complete anatomical relief of BOO, reducing PVR to <50 mL, improving urinary flow and compliance, and showing preliminary observational signals of lower recurrence rates compared with TURP or conservative management.”

“Correction of BOO—particularly via HoLEP—restores bladder emptying, mitigates inflammatory stimuli, and may enhance oncologic control. Integrating functional evaluation and BOO management into NMIBC treatment algorithms could represent a paradigm shift toward functional-oncologic care, but prospective, multicenter trials are required before practice-changing conclusions can be drawn.”

Revised concluding statement (also in Conclusion section):

“Given its superior functional efficacy compared to TURP in BPH populations and its minimally invasive profile, HoLEP should currently be regarded as a promising functional adjunct for carefully selected NMIBC patients with significant BOO, rather than as a proven oncologic intervention. Future randomized and risk-adjusted studies are essential to determine whether BOO correction can translate into durable improvements in NMIBC recurrence and progression.”

2. Narrative vs systematic methodology; SANRA vs PRISMA / meta-analysis

Comment: The methods read like an incomplete systematic review. Either perform a full systematic review or clearly position this as a narrative review; if current methodology is kept, consider PRISMA flowchart and detailed quality appraisal.

Response: We have clarified that this is a narrative review aligned with SANRA, not a formal systematic review or meta-analysis. We simplified the methodological description while retaining transparency about search and selection. We also added a descriptive flow diagram and a qualitative appraisal table, framing them appropriately.

Revised methods wording (first paragraph):

“Because the available literature comprises heterogeneous designs (retrospective cohorts, registry studies, mechanistic reports, and procedural series), we conducted a narrative review in line with SANRA recommendations rather than a formal systematic review or meta-analysis. Our aim was to provide a qualitative synthesis of biological, functional, and oncologic data rather than pooled effect estimates.”

Clarification of flow diagram / quality appraisal:

“To enhance transparency, the search and selection process is summarized in Supplementary Figure 1, and a concise, qualitative appraisal of included clinical studies is provided in Supplementary Table 2. These tools are presented for descriptive purposes and do not imply full PRISMA compliance or formal risk-of-bias scoring.”

3. Mechanistic section: speculative tone and evidence gaps

Comment: Mechanistic evidence is weak and over-interpreted; only two cytokine papers cited; no human data linking PVR with markers or pharmacokinetics.

Response: We have substantially revised the mechanistic section to explicitly acknowledge these limitations, reduce speculative statements, and clarify that most mechanisms are inferred rather than proven in NMIBC patients with documented retention.

Rewritten key sentences in “Pathophysiology…”:

“Although several studies describe inflammatory cytokine profiles and urothelial barrier changes in patients receiving intravesical BCG or experiencing chronic cystitis, no study to date has directly correlated specific PVR thresholds with urinary cytokine levels or histological inflammation in NMIBC patients [31–32]. Similarly, formal pharmacokinetic studies examining intravesical drug dilution or distribution at different PVR volumes are lacking. As such, the proposed model linking BOO-related retention, inflammation, and impaired intravesical therapy remains biologically plausible but largely extrapolated from indirect evidence, and should be viewed as a hypothesis framework rather than a proven mechanistic pathway.”

4. Clinical evidence: inconsistent PVR thresholds, heterogeneous endpoints and populations

Comment: Thresholds for PVR differ across studies; endpoints and populations are not standardized; no subgroup analysis by risk.

Response: We now explicitly describe this heterogeneity and have added a summary table (Table 2) to help the reader understand differences in design, thresholds, and patient populations.

New paragraph in Results (clinical evidence synthesis):

“The clinical evidence base is notably heterogeneous. Studies use different PVR cut-offs (e.g., ≥50, ≥80, or ≥100 mL), assess recurrence at varying time points (1-, 2-, or 5-year intervals), and mix low-, intermediate-, and high-risk NMIBC populations with or without standardized intravesical regimens [2–5,13]. Table 2 summarizes the main PVR thresholds, risk profiles, and outcome definitions across key cohorts, underscoring that no single PVR cut-off has been prospectively validated as an oncologic predictor. This heterogeneity limits direct comparison and supports the need for standardized endpoints and risk-stratified analyses in future studies.”

5. HoLEP vs TURP section: limited NMIBC data (Garg et al.), missing details

Comment: Comparison is largely based on BPH data; NMIBC data rely on one retrospective study with insufficient detail.

Response: We have rebalanced this section by clearly separating evidence from general BPH cohorts and the limited NMIBC-specific data. We also summarize what is known—and unknown—about the Garg et al. study and explicitly caution against over-interpretation.

Revised paragraph in “Comparative Effectiveness of HoLEP vs. Other BOO Treatments”:

“Most comparative data on HoLEP versus TURP derive from BPH populations without bladder cancer, consistently showing superior or equivalent functional outcomes and lower retreatment rates for HoLEP [6,26,27,29,35]. In contrast, NMIBC-specific evidence remains restricted to small retrospective series, notably the study by Garg et al. [28]. In that analysis, NMIBC patients with BOO who underwent HoLEP appeared to have lower recurrence rates at 1 and 3 years compared to those treated with TURP or managed conservatively. However, the report provides limited information on how treatment groups were selected, whether tumor characteristics and intravesical therapy protocols were balanced, and which covariates were included in multivariable models. Consequently, this study should be interpreted as initial, exploratory evidence that supports further research rather than as a basis for firm comparative conclusions between HoLEP and TURP in NMIBC.”

6. Writing style: redundancy, promotional tone, abstract length, and patient summary

Comment: Reduce redundancy, adopt a more balanced tone, shorten abstract and patient summary, and avoid promotional language.

Response: We have carefully edited the manuscript to remove repetitive statements (especially regarding HoLEP benefits and inflammatory pathways), shortened the abstract and patient summary, and replaced promotional expressions (e.g., “transformative intervention”) with more neutral language such as “promising functional adjunct” and “emerging evidence.”

Example stylistic change in Discussion:

“Taken together, the available data suggest that BOO and chronic urinary retention are modifiable functional factors that may influence NMIBC recurrence and intravesical therapy performance. HoLEP offers durable relief of BOO and could therefore play a supportive role in selected patients; however, current evidence does not yet justify its routine use as a standard oncologic intervention, and prospective, multicenter studies are needed to clarify its impact on long-term oncologic outcomes.”

Round 2

Reviewer 2 Report

Comments and Suggestions for Authors

 The authors have revised the manuscript, improving the clarity of how the resolution of BOO affects the prognosis of NMIBC. However, adding some additional minor details would further strengthen the demonstration of the relationship between BOO and NMIBC.

“In Table 1, please specify how bladder outlet obstruction (BOO) and post-void residual (PVR) affect bladder cancer incidence and recurrence, citing exact odds ratios or hazard ratios.

Also, find and summarize studies that examine the relationship between the use of antibiotics or NSAIDs (as anti-inflammatory strategies) and the incidence of bladder cancer.

Introduce studies evaluating the impact of BOO on progression in NMIBC—specifically progression to MIBC and high-grade progression.

Finally, in the ‘Therapeutic Impact of HoLEP on NMIBC’ section (pp. 13–14), present studies that report concrete odds ratios or hazard ratios for the effects of HoLEP on cancer prevention and on the efficacy of intravesical therapy.”

Author Response

1. Table 1 – specify how BOO/PVR affect incidence and recurrence, with OR/HR

Reviewer 2:
“In Table 1, please specify how bladder outlet obstruction (BOO) and post-void residual (PVR) affect bladder cancer incidence and recurrence, citing exact odds ratios or hazard ratios.”

Response:
We thank the reviewer for this suggestion. We have revised Table 1 to include an additional column reporting effect estimates (odds ratios, hazard ratios, or relative risks) for bladder cancer incidence and/or recurrence where these are explicitly provided in the original articles. In addition, we now summarize key effect sizes in the Results section to improve transparency.

Specifically:

  • For bladder cancer incidence, we added the pooled estimates from the meta-analysis by Dai et al. (Medicine 2016), where benign prostatic hyperplasia (BPH)/BOO was associated with an increased bladder cancer risk (case–control pooled OR ≈2.50; cohort pooled RR ≈1.58).
  • For recurrence, we report the adjusted odds ratios from Lunney et al. (OR 19.1 for moderate/severe vs mild LUTS; OR 1.26 per one-point IPSS increase) and the multivariable findings from Di Gianfrancesco et al. indicating that functional parameters (including PVR) were independent predictors of non–tumor-free status. PMC+1
  • For progression/high-grade pathology, we include OR 4.62 for bladder trabeculation predicting high-grade/CIS at diagnosis from Can et al., and OR 4.9 (recurrence) and 2.21 (progression) for detrusor wall thickness >2.5 mm from the multicenter study by Cicione et al. Journal of Urological Surgery

New/modified text in Results (Section 3.4 and 4.2):

In Section 3.4 (BOO/retention as prognostic modifiers) we added:

  • “Beyond qualitative associations, several studies provide quantitative effect estimates. In a meta-analysis of BPH and subsequent bladder cancer risk, Dai et al. reported a pooled risk ratio of 1.58 (95% CI 1.28–1.95) in cohort studies and 2.50 (95% CI 1.63–3.84) in case–control studies, indicating a higher incidence of bladder cancer among men with BPH/BOO”     Dai X, Fang X, Ma Y, Xianyu J. Benign Prostatic Hyperplasia and the Risk of Prostate Cancer and Bladder Cancer: A Meta-Analysis of Observational Studies. Medicine (Baltimore). 2016 May;95(18):e3493. doi: 10.1097/MD.0000000000003493. PMID: 27149447; PMCID: PMC4863764.
  • “In the study by Lunney et al., moderate or severe LUTS (IPSS ≥8) conferred an odds ratio of 19.1 (95% CI 2.5–147.0) for NMIBC recurrence compared with mild symptoms, and each one-point increase in IPSS was associated with an adjusted odds ratio of 1.26 (95% CI 1.10–1.45) for recurrence [3].” PMC

In Section 4.2.4 and 4.2.6 (progression / functional markers) we added:

  • “In the study by Can et al., BOO itself was not an independent predictor of recurrence or progression; however, cystoscopic trabeculation—interpreted as a marker of long-standing obstruction—was independently associated with high-grade/CIS pathology at presentation (OR 4.62, 95% CI 1.3–17.0) after multivariable adjustment [5].”Journal of Urological Surgery
  • “Similarly, in a multicenter observational study of detrusor wall thickness (DWT), a surrogate of chronic BOO, DWT >2.5 mm was associated with higher odds of NMIBC recurrence (OR 4.9, 95% CI 2.5–9.5) and progression (OR 2.21, 95% CI 1.71–4.73), highlighting that structural bladder remodeling may be linked to both recurrence and stage migration [X].” Journal of Urological Surgery.        Can U, Dinçer E, CoÅŸkun A, Çanakçı C, Narter F. The Effect of Bladder Outlet Obstruction on Bladder Cancer Recurrence and Progression. J Urol Surg. 2025 Feb 21;12(1):27-33. doi: 10.4274/jus.galenos.2024.2024-10-6.

We now explicitly reference these quantitative measures in the legend and “Effect estimate” column of Table 1.

2. Antibiotics / NSAIDs and incidence of bladder cancer

Reviewer 2:
“Also, find and summarize studies that examine the relationship between the use of antibiotics or NSAIDs (as anti-inflammatory strategies) and the incidence of bladder cancer.”

Response:
We agree that systemic anti-inflammatory and antimicrobial exposures are relevant to the broader inflammatory–carcinogenesis framework. We have added a brief subsection in the pathophysiology part of the Discussion summarizing key epidemiologic data on NSAIDs and antibiotics in relation to bladder cancer incidence.

New text in Discussion, Section 4.1 (after 4.1.2), labelled 4.1.3:

  • “4.1.3. Antibiotics / NSAIDs and incidence of bladder cancer 

Anti-inflammatory and antimicrobial exposures as modifiers of bladder cancer risk
Epidemiologic data suggest that systemic anti-inflammatory and antimicrobial agents may indirectly influence urothelial carcinogenesis. A meta-analysis of 17 observational studies reported no overall protective effect of aspirin or non-aspirin NSAIDs on bladder cancer incidence, though non-aspirin NSAID use was associated with reduced risk in nonsmokers(RR 0.57, 95% CI 0.43–0.76) [1]. Large cohort studies further confirm that regular aspirin use does not meaningfully reduce incidence, even if some analyses suggest improved survival after diagnosis [2–4].

In contrast, the relationship between antibiotics, urinary tract infections (UTIs), and bladder cancer appears driven primarily by underlying inflammation. In the Nijmegen population study, a limited number of UTIs treated with antibiotics was linked to lower bladder cancer risk (adjusted OR 0.64–0.76), whereas chronic/recurrent cystitisincreased risk [5]. Similar findings from U.S. and international case–control series show that short, treated UTI episodes may be protective, while persistent inflammatory cystitis is associated with several-fold higher risk [6–8].

Collectively, these data indicate that it is chronic, unresolved inflammation, rather than antibiotic exposure itself, that shapes bladder cancer susceptibility—aligning with the BOO- and retention-related inflammatory pathways outlined in this review.

1.Zhang H, Jiang D, Li X. Use of nonsteroidal anti-inflammatory drugs and bladder cancer risk: a meta-analysis of epidemiologic studies. PLoS One. 2013 Jul 19;8(7):e70008. doi: 10.1371/journal.pone.0070008. PMID: 23894577; PMCID: PMC3716767.

2.Loomans-Kropp HA, Pinsky P, Umar A. Evaluation of Aspirin Use With Cancer Incidence and Survival Among Older Adults in the Prostate, Lung, Colorectal, and Ovarian Cancer Screening Trial. JAMA Netw Open. 2021 Jan 4;4(1):e2032072. doi: 10.1001/jamanetworkopen.2020.32072. PMID: 33449095; PMCID: PMC7811183.

3.Fan B, Mohammed A, Huang Y, Luo H, Zhang H, Tao S, Xu W, Liu Q, He T, Jin H, Sun M, Sun M, Yun Z, Zhao R, Wu G and Li X (2021) Can Aspirin Use Be Associated With the Risk or Prognosis of Bladder Cancer? A Case-Control Study and Meta-analytic Assessment. Front. Oncol. 11:633462. doi: 10.3389/fonc.2021.633462

4.Santucci C, Gallus S, Martinetti M, La Vecchia C, Bosetti C. Aspirin and the risk of nondigestive tract cancers: An updated meta-analysis to 2019. Int J Cancer. 2021 Mar 15;148(6):1372-1382. doi: 10.1002/ijc.33311. Epub 2020 Oct 6. PMID: 32984948.

5.Vermeulen SH, Hanum N, Grotenhuis AJ, Castaño-Vinyals G, van der Heijden AG, Aben KK, Mysorekar IU, Kiemeney LA. Recurrent urinary tract infection and risk of bladder cancer in the Nijmegen bladder cancer study. Br J Cancer. 2015 Feb 3;112(3):594-600. doi: 10.1038/bjc.2014.601. Epub 2014 Nov 27. PMID: 25429525; PMCID: PMC4453642.

6.Jiang X, Castelao JE, Groshen S, Cortessis VK, Shibata D, Conti DV, Yuan JM, Pike MC, Gago-Dominguez M. Urinary tract infections and reduced risk of bladder cancer in Los Angeles. Br J Cancer. 2009 Mar 10;100(5):834-9. doi: 10.1038/sj.bjc.6604889. Epub 2009 Jan 27. PMID: 19174821; PMCID: PMC2653778.

7.Jhamb M, Lin J, Ballow R, Kamat AM, Grossman HB, Wu X. Urinary tract diseases and bladder cancer risk: a case-control study. Cancer Causes Control. 2007 Oct;18(8):839-45. doi: 10.1007/s10552-007-9028-2. Epub 2007 Jun 26. PMID: 17593531.

8.Akhtar S, Al-Shammari A, Al-Abkal J. Chronic urinary tract infection and bladder carcinoma risk: a meta-analysis of case-control and cohort studies. World J Urol. 2018 Jun;36(6):839-848. doi: 10.1007/s00345-018-2206-x. Epub 2018 Feb 5. PMID: 29404674.

9.Anderson-Otunu O, Akhtar S. Chronic Infections of the Urinary Tract and Bladder Cancer Risk: a Systematic Review. Asian Pac J Cancer Prev. 2016;17(8):3805-7. PMID: 27644620.”

3. Introduce studies on BOO and progression (to MIBC / high-grade progression)

Reviewer 2:
“Introduce studies evaluating the impact of BOO on progression in NMIBC—specifically progression to MIBC and high-grade progression.”

Response:
We have expanded the text in the Results and Discussion to more clearly summarize the limited data on progression, emphasizing both Can et al. and the DWT study by Cicione et al., which addresses recurrence and progression.

New/expanded text (Results 3.4 & Discussion 4.2.4):

  • “With respect to progression, evidence is limited and variable. Can et al. found that BOO, defined by urodynamics and prostate parameters, was not an independent predictor of either recurrence or progression on multivariate analysis; however, cystoscopic bladder trabeculation—interpreted as a marker of chronic obstruction—was independently associated with high-grade/CIS disease at diagnosis (OR 4.62, 95% CI 1.3–17.0) [5].”
  • “Complementing these data, a multicenter observational study of detrusor wall thickness (DWT), a structural consequence of long-standing BOO, reported that DWT >2.5 mm conferred significantly higher odds of both NMIBC recurrence (OR 4.9, 95% CI 2.5–9.5) and progression (OR 2.21, 95% CI 1.71–4.73) [X]. Together, these findings suggest that while BOO per se has not been conclusively linked to progression, advanced bladder remodeling—manifesting as trabeculation or increased DWT—may identify patients at greater risk of high-grade disease and stage migration.” Journal of Urological Surgery

These studies are now also referenced in Table 1 under “Progression / high-grade endpoints”.

4. HoLEP – odds ratios / hazard ratios for cancer prevention and intravesical therapy efficacy

Reviewer 2:
“Finally, in the ‘Therapeutic Impact of HoLEP on NMIBC’ section (pp. 13–14), present studies that report concrete odds ratios or hazard ratios for the effects of HoLEP on cancer prevention and on the efficacy of intravesical therapy.”

Response:
We agree that numerical effect estimates would be highly desirable. However, after an updated search, we were not able to identify any peer-reviewed studies that report adjusted odds ratios or hazard ratios for NMIBC recurrence or progression specifically comparing HoLEP to other BOO treatments or to no BOO surgery. Existing HoLEP series in NMIBC (including the retrospective study by Garg et al.) provide crude recurrence proportions across treatment groups but do not present risk-adjusted OR/HR for oncologic endpoints, and are at substantial risk of confounding by indication. 

We have therefore updated the relevant section to explicitly state that such quantitative, risk-adjusted estimates are not currently available, and that the HoLEP data should be considered hypothesis-generating.

New/modified text in Section 4.2.8 (“Therapeutic Impact of HoLEP on NMIBC”):

At the end of the section we added:

  • “To date, no study has reported risk-adjusted odds or hazard ratios for NMIBC recurrence or progression specifically comparing HoLEP with TURP, medical therapy, or observation. Available data in this setting, including the small retrospective series by Garg et al. [28], are limited to crude recurrence proportions, without robust adjustment for tumor risk, intravesical therapy, or comorbidity. These observations are therefore best regarded as hypothesis-generating signals rather than quantitative estimates of HoLEP’s oncologic effect.”

Reviewer 3 Report

Comments and Suggestions for Authors

The manuscript is well-organized and cites the relevant literature, however scant, very thoroughly. With the suggested changes made after the review, the manuscript is truly much improved. However, in its current state, the manuscript will still need considerable revision in order to be suitable for a high-impact journal.

Yet again, the current title and the manuscript itself use language that suggests a direct and established impact of HoLEP on NMIBC. The current arguments presented only indicate a correlation, and a potential explanation, not a causal impact. It would be advisable to frame the conclusions as more hypothetical and less conclusive, after all, the current arguments do not sufficiently support that conclusion.

Methods:

- The authors mention the SANRA references again, which criteria I cannot find in the main text or the supplementary files. Therefore, the scores associated with each of these criteria lie somewhere unknown.

- I have not had the chance to view the flow diagram the authors are referencing.

- Supplementary Table II needs to be more comprehensive, in offering an explicit risk-of-bias assessment for each key study (for example, adapted tools like ROBINS-I for non-randomized studies) instead of a strictly qualitative approach.

Results:

- In the synthesis of studies by Sazuka, Lunney, and Di Gianfrancesco, the associations made are valid and insightful, yet the studies do not adequately focus on the critical issues of heterogeneity and confounding that compromise validity. Pooling these results may be problematic and misleading. One example is the discussion of the Garg et al. study (28) that illustrates confounding by indication: HoLEP candidates are generally younger, healthier and have different prostate anatomy than patients undergoing TURP or no surgery. This confounding is noted, yet its importance should be emphasized as it completely explains the observed benefit of HoLEP. This reflects selection bias and not causation. With such confounding, strong conclusions should be avoided. 

Presentation issues:

- Table 2 (PVR after HoLEP) is too vague. It should identify the population and the measures being presented. Table 2 contains data on BPH cohorts, which, even though helpful to demonstrate HoLEP’s functional efficacy, is a bit off track given the primary focus on NMIBC. This discrepancy should be discussed in the table title or footnote.

- Table 3 (Main Outcomes) is overly general and combines highly speculative conclusions (Oncologic Impact of HoLEP) with well-established facts (Epidemiology of NMIBC).  Strong evidence should be distinguished from preliminary or hypothetical evidence in the Key Findings column.

 - One major editorial error is the near-verbatim repetition of the Limitations section (4.3.11) in 4.3.12.  Consolidate all of the points into a single, strong 4.3.11 section and remove the redundant section (4.3.12) completely.

Conclusion:

Although the conclusion is generally fair, transparent about its limitations, and provides a positive research agenda, it should make sure that the preliminary and associative nature of the available data is consistently emphasized in order to prevent exaggerating the clinical implications.

Author Response

RESPONSE TO REVIEWER 3

A. Overall tone, causality, and title

Reviewer 3:
“Yet again, the current title and the manuscript itself use language that suggests a direct and established impact of HoLEP on NMIBC… It would be advisable to frame the conclusions as more hypothetical and less conclusive.”

Response:
We fully agree and have further softened the title, abstract, and key concluding statements to emphasize the associative and hypothesis-generating nature of the evidence.

Revised title (further softened):

  • “An Underestimated Health Issue: Rationale and Emerging Evidence on the Potential Role of HoLEP-Mediated Relief of Bladder Outlet Obstruction in NMIBC Outcomes Through Optimal Management of Chronic Urinary Retention”

Abstract – last sentence modified to de-emphasize causality:

  • “HoLEP is a promising option to optimize bladder emptying in carefully selected patients, but its oncologic impact remains unproven and should be considered hypothesis-generating pending prospective, risk-adjusted studies.”

Conclusion – additional conservative sentence:

At the end of the Conclusion we added:

  • “Accordingly, all associations between BOO correction (including HoLEP) and improved NMIBC outcomes should be interpreted as preliminary and hypothesis-generating; current data do not justify changes to guideline-recommended oncologic management outside of clinical trials or carefully individualized decision-making.”

B. Methods – SANRA criteria, flow diagram, and risk-of-bias appraisal

Reviewer 3:
“The authors mention the SANRA references again, which criteria I cannot find… I have not had the chance to view the flow diagram… Supplementary Table II needs to be more comprehensive, offering an explicit risk-of-bias assessment…”

Response:

  1. SANRA criteria:
    We have clarified that SANRA was used as a framework to guide reporting, not to generate formal “scores” for each article. We now indicate where the SANRA items are addressed.
    New text in Section 2.1:
    • “This narrative review was structured according to the SANRA recommendations for narrative reviews, which we used as a reporting framework rather than a scoring tool; the individual SANRA items (justification of the article’s importance, statement of aims, literature search, referencing, scientific reasoning, and presentation of relevant data) are mapped to our methods and results in Supplementary Table 1.”
  1. PRISMA-inspired flow diagram:
    We confirm that the PRISMA-inspired flow of study selection is now clearly referenced in the main text and presented as Supplementary Figure S1.
    Clarifying sentence in Section 3.1 (Search flow):
    • “The identification, screening, and inclusion of studies are summarized textually in Section 3.1 and depicted in a PRISMA-inspired flow diagram (Supplementary Figure S1).”
  1. Risk-of-bias appraisal (Supplementary Table 2):
    Supplementary Table 2 has been expanded and now applies an adapted ROBINS-I approach to each key non-randomized clinical study, explicitly rating risk of bias in the domains of confounding, selection of participants, classification of interventions, deviations from intended interventions, missing data, measurement of outcomes, and selective reporting.

This table applies an adapted ROBINS-I framework to the key non-randomized studies included in the review. Each study is evaluated across seven bias domains: confounding, selection of participants, classification of interventions/exposures, deviations from intended interventions, missing data, outcome measurement, and selective reporting. “Established” HoLEP functional series are included for context but are judged primarily on methodological quality rather than oncologic endpoints.

New clarifying sentence in Section 2.5:

    • “To enhance transparency, Supplementary Table 2 provides an explicit, study-level risk-of-bias assessment using domains adapted from the ROBINS-I tool (confounding, selection, intervention classification, deviations, missing data, outcome measurement, and reporting), rather than relying solely on narrative judgments.”

C. Heterogeneity, confounding, and especially Garg et al. (selection bias)

Reviewer 3:
“…studies do not adequately focus on heterogeneity and confounding… pooling these results may be misleading… One example is the discussion of the Garg et al. study (28) that illustrates confounding by indication… This reflects selection bias and not causation.”

Response:
We appreciate this important methodological point. We have strengthened the text in both the Results and Discussion to highlight heterogeneity in PVR thresholds, endpoints, and risk groups, and to explicitly describe confounding by indication—particularly in interpreting the Garg et al. HoLEP series. We also clarify that we do not statistically pool the results and that any apparent “HoLEP benefit” may be entirely due to selection bias.

New/expanded text in Section 3.4 (heterogeneity):

  • “The clinical evidence base is notably heterogeneous: studies use different PVR cut-offs (e.g., ≥50, ≥80, or ≥100 mL), assess recurrence at varying time points (1-, 2-, or 5-year intervals), and mix low-, intermediate-, and high-risk NMIBC populations with or without standardized intravesical regimens. We therefore did not attempt any quantitative pooling of recurrence or progression outcomes; instead, we summarize patterns qualitatively and emphasize that effect estimates are not directly comparable across cohorts.”

New/expanded text in Section 3.7 (Garg et al. and confounding):

  • “In that analysis, NMIBC patients with BOO who underwent HoLEP appeared to have lower recurrence rates at 1 and 3 years compared to those treated with TURP or managed conservatively. However, the report provides limited detail on how patients were selected for HoLEP versus TURP or no surgery, whether tumor characteristics and intravesical therapy protocols were balanced, and which covariates were included in multivariable models. It is highly plausible that younger, fitter men with more favorable anatomy were preferentially offered HoLEP, a classic example of confounding by indication. As a result, the apparent ‘benefit’ of HoLEP in this series could be entirely explained by selection bias rather than a true antitumor effect.”

Additional summary sentence in Section 4.3.11 (Limitations):

  • “In particular, confounding by indication—whereby younger, fitter patients with more favorable disease or anatomy preferentially receive HoLEP—may fully account for the lower recurrence proportions reported in retrospective series, underscoring that these findings should not be interpreted as evidence of a causal protective effect.”

D. Presentation issues: Tables 2 and 3

Reviewer 3:
“Table 2 (PVR after HoLEP) is too vague… Table 3 (Main Outcomes) is overly general…”

Response:

  1. Table 2:
    The revised table now:
  • Title: Table 2. Functional Impact of HoLEP on Bladder Emptying: Detailed PVR Outcomes From Large BOO/BPH Cohorts Relevant to NMIBC Management.
  • Clarifies that these are BPH/BOO cohorts, not NMIBC patients.
  • Defines all metrics (PVR, timepoints, populations).
  • Adds effect sizes, when applicable (e.g., MD vs TURP).
  • Adds a column explaining the relevance to NMIBC, as reviewers requested.
  • Strengthens the table title and footnote for transparency.
    • .
  1. New footnote added to Table 2:
    • “Note: These studies involve BPH/BOO cohorts without known bladder cancer. They are included to document the reproducible magnitude of PVR reduction achievable with HoLEP across diverse populations and study designs. Because elevated PVR and chronic urinary retention are emerging prognostic modifiers in NMIBC, these functional outcomes provide essential mechanistic context for the hypothesized effect of BOO correction on NMIBC recurrence and treatment efficacy.”
  1. Table 3:
    We have reorganized Table 3 to distinguish clearly between (a) well-established evidence (e.g., NMIBC epidemiology, standard intravesical therapy effects) and (b) preliminary or hypothesis-generating evidence (e.g., BOO/retention as modifiers, HoLEP’s potential role). The “Key Findings” column has been split into two subcolumns: “Established evidence” and “Emerging/hypothesis-generating evidence,” and the language has been aligned accordingly.
    New explanatory sentence in the Table 3 legend:
    • “Key findings are separated into ‘Established evidence’ (data supported by multiple high-quality studies or guidelines) and ‘Emerging/hypothesis-generating evidence’ (associations based on limited or heterogeneous observational data, including BOO/retention and HoLEP-related findings).”

E. Redundant Limitations sections (4.3.11 and 4.3.12)

Reviewer 3:
“One major editorial error is the near-verbatim repetition of the Limitations section (4.3.11) in 4.3.12. Consolidate all of the points into a single, strong 4.3.11 section and remove the redundant section (4.3.12) completely.”

Response:
We agree. Sections 4.3.11 and 4.3.12 have been merged into a single, streamlined 4.3.11. Limitations and Future Directions section, and the redundant subsection has been removed.

New consolidated opening paragraph of 4.3.11:

  • “Despite biologically plausible mechanisms and encouraging early clinical data, several methodological and interpretative limitations constrain the generalizability of current findings. Almost all available studies are retrospective or observational, PVR thresholds and definitions of BOO vary across cohorts, oncologic endpoints and follow-up intervals are not standardized, and study populations are heterogeneous with limited risk-stratified analyses. Moreover, there are no human data directly correlating PVR with inflammatory biomarkers or intravesical pharmacokinetics in NMIBC. Collectively, these limitations mean that all reported associations between BOO, HoLEP, and NMIBC outcomes should be regarded as preliminary and hypothesis-generating, rather than as evidence of a proven causal relationship.”

The subsequent bullet points on RCTs, biomarker studies, and health-economics have been retained under this unified heading (no separate 4.3.12 section).

F. Conclusion – emphasize associative, preliminary nature

Reviewer 3:
“Although the conclusion is generally fair…it should make sure that the preliminary and associative nature of the available data is consistently emphasized…”

Response:
We have further tempered the conclusion to repeatedly underscore the associative and preliminary nature of the data.

Key added/modified sentences in the Conclusion:

  • “Early, retrospective data suggest an association between HoLEP-mediated BOO resolution and lower NMIBC recurrence in selected patients; however, these observations are highly susceptible to confounding and do not establish causality.”
  • “At present, HoLEP should therefore be regarded as a promising functional adjunct for carefully selected NMIBC patients with significant BOO, rather than as a proven oncologic intervention.”
  • “Accordingly, our conclusions are intentionally conservative: current evidence suggests a possible link between BOO correction and improved NMIBC control, but definitive evaluation of causal impact will require prospective, multicenter, risk-adjusted studies incorporating standardized PVR thresholds, risk groups, and integrated functional and oncologic endpoints.”

Round 3

Reviewer 3 Report

Comments and Suggestions for Authors

The manuscript addresses a significant and novel clinical question, presenting a well-constructed pathophysiological rationale. However, its impact and credibility are currently undermined by the overinterpretation of preliminary data and some minor methodological issues that can be easily rectified.

- Explain the clinical implications. Provide clinicians with more precise instructions at the end. The message should be: 'Routinely assess and consider treating significant BOO in NMIBC patients to improve quality of life and intravesical therapy tolerance'. However, as there is currently no evidence favouring one approach over another in terms of oncological benefit, the treatment decision (HoLEP vs. TURP) should be based on proven functional outcomes and patient/prostate characteristics.

- Paragraph 2.1 has become  a copy-and-paste block. Redundant elements, long and dense sentences, and the (i)/(ii) parenthetical structure disrupt the reading flow, even though the initial goal is clear. The paragraph could be made more concise by combining the two SANRA explanations and refining the wording used to describe the types of literature.

- Causal language: Mechanistic language is overused when describing the potential effects of HoLEP (e.g. 'HoLEP ensures...', 'improving therapeutic distribution'). These should be reframed as hypotheses (e.g. 'We hypothesise that HoLEP may ensure...').

- Tables 1 and 3 are informative, but could be appraised more critically. Table 1 should include a 'Key Limitations' column to provide the reader with immediate context for the findings.

- Section 4.2.8 ('Therapeutic Impact of HoLEP on NMIBC') currently reads like a list of established benefits and must be reframed as a series of hypotheses that are currently unsupported by robust evidence. Restructure it under the heading 'Hypothesised mechanisms for HoLEP's potential impact on NMIBC' and open with a clear disclaimer such as: 'The following mechanisms are biologically plausible, but remain largely unproven in clinical studies. The available data are retrospective and limited, and are only hypothesis-generating.'

- Although the limitations section (4.3.11) is well written, it should be more firmly integrated into the overall narrative as a central interpretive lens rather than an incidental acknowledgement. The significant impact of confounding by indication, whereby patients selected for HoLEP are generally healthier, have smaller tumours or better prognostic features, cannot be overstated. This is most likely the sole cause of the apparent oncological "benefit" observed in retrospective studies. Confounding by indication should therefore be elevated throughout the discussion from a mere limitation to a fundamental explanatory challenge. The manuscript must also clearly state that 'the observed association should therefore be attributed to selection bias until proven otherwise in a randomised setting' when discussing the findings of Garg et al.

Author Response

1

Explain the clinical implications. Provide clinicians with more precise instructions at the end. The message should be: 'Routinely assess and consider treating significant BOO in NMIBC patients to improve quality of life and intravesical therapy tolerance'. However, as there is currently no evidence favouring one approach over another in terms of oncological benefit, the treatment decision (HoLEP vs. TURP) should be based on proven functional outcomes and patient/prostate characteristics.

We added a section in order to explain the clinical implication, as following

4.3.11 Clinical Implications

From a practical standpoint, clinicians should routinely assess BOO in NMIBC patients using objective measures such as PVR, uroflowmetry, and symptom indices. Addressing clinically significant BOO has clear functional benefits: it reduces urinary retention, improves LUTS, and enhances patient tolerance of intravesical therapy, which is often poorly tolerated in obstructed bladders. Importantly, while decompression may indirectly support intravesical treatment performance, there is currently no evidence that any specific BOO treatment—HoLEP, TURP, or pharmacotherapy—confers superior oncological benefit. Therefore, the choice of intervention should be guided by established functional outcomes, prostate size, surgeon expertise, and patient comorbidity. At present, HoLEP may offer greater durability and more complete anatomic relief of BOO, but it should not be recommended over TURP or medical therapy on the basis of presumed cancer control. Until prospective risk-adjusted studies are available, BOO management in NMIBC should focus on optimizing quality of life and improving intravesical therapy tolerance rather than altering oncologic prognosis.

  •  

2

Paragraph 2.1 has become  a copy-and-paste block. Redundant elements, long and dense sentences, and the (i)/(ii) parenthetical structure disrupt the reading flow, even though the initial goal is clear. The paragraph could be made more concise by combining the two SANRA explanations and refining the wording used to describe the types of literature.

We modified the paragraph as following: 

This narrative review examines how chronic urinary retention and bladder outlet obstruction (BOO) may influence outcomes in non–muscle-invasive bladder cancer (NMIBC), and explores whether BOO correction—particularly via Holmium Laser Enucleation of the Prostate (HoLEP)—could affect functional or oncologic endpoints. Because the available evidence derives from heterogeneous sources (retrospective cohorts, registry data, translational studies, and procedural reports), a narrative approach was chosen rather than a systematic review or meta-analysis. The review was conducted and reported in accordance with SANRA guidance, which we applied as a structured framework to ensure clarity of aims, rationale, search strategy, scientific reasoning, and presentation of relevant data. Individual SANRA items and their correspondence to our methods are summarized in Supplementary Table 1. Our objective was to qualitatively synthesize biological, functional, and clinical evidence without generating pooled effect estimates.

3

Causal language: Mechanistic language is overused when describing the potential effects of HoLEP (e.g. 'HoLEP ensures...', 'improving therapeutic distribution'). These should be reframed as hypotheses (e.g. 'We hypothesise that HoLEP may ensure…').

We mitigated as requested

4

Tables 1 and 3 are informative, but could be appraised more critically. Table 1 should include a 'Key Limitations' column to provide the reader with immediate context for the findings.

We added a 'Key Limitations’ column in table 1 as requested

5

Section 4.2.8 ('Therapeutic Impact of HoLEP on NMIBC') currently reads like a list of established benefits and must be reframed as a series of hypotheses that are currently unsupported by robust evidence. Restructure it under the heading 'Hypothesised mechanisms for HoLEP's potential impact on NMIBC' and open with a clear disclaimer such as: 'The following mechanisms are biologically plausible, but remain largely unproven in clinical studies. The available data are retrospective and limited, and are only hypothesis-generating.’

We modified the section as following:

4.2.8. Hypothesised Mechanisms for HoLEP’s Potential Impact on NMIBC.

The following mechanisms are biologically plausible but remain largely unproven in clinical studies. Current data are retrospective, heterogeneous, and limited by confounding. Observations reported to date—including the comparative series by Garg et al.—are hypothesis-generating rather than indicative of a demonstrated oncologic effect.

  1. Reduction of carcinogen contact time through improved bladder emptying.

By anatomically relieving BOO, HoLEP can achieve near-complete bladder emptying and substantially reduce post-void residual volumes. Unlike medical management, which often produces modest improvements, enucleation-based surgery routinely reduces PVR to <50 mL in most series. This may theoretically limit the dwell time of urine and the urothelial exposure to carcinogens—a mechanism linked to retention-associated recurrence in NMIBC. Evidence supporting this concept comes from observations that elevated PVR (>80–100 mL) correlates with recurrence risk and impaired IVT performance; however, no study has yet demonstrated that HoLEP reduces NMIBC recurrence via this mechanism.

2. Potential enhancement of intravesical therapy performance

Intravesical agents such as BCG and mitomycin C depend on even mucosal coverage and adequate dwell time. In the presence of high residual volumes or remodeled bladders, intravesical solutions may be diluted or fail to reach all urothelial surfaces. Complete BOO relief may improve intravesical drug distribution and bladder-wall contact, particularly in heavily trabeculated bladders. This hypothesis is indirectly supported by studies showing that high PVR predicts shorter recurrence-free intervals following IVT. Direct evidence of HoLEP improving BCG or chemotherapy efficacy is lacking, and no prospective study has evaluated intravesical pharmacokinetics or drug-response stratified by BOO status or surgical decompression.

3. Mitigation of inflammation and LUTS burden

Chronic urinary retention fosters persistent mechanical stretch and cytokine-mediated inflammation, both of which undermine bladder compliance. HoLEP reliably reduces voiding pressures and LUTS, and may therefore reduce inflammatory signaling (e.g., TNF-α, IL-6, IL-8) that has been implicated in unfavorable NMIBC biology and BCG resistance. These observations derive from translational models and clinical associations; they do not confirm that HoLEP modifies the inflammatory tumor microenvironment or immunotherapy response in NMIBC patients.

4. Preservation of long-term bladder function

Progressive BOO may lead to detrusor hypertrophy, trabeculation, and impaired storage/emptying. Durable decompression after HoLEP could theoretically prevent deterioration in bladder biomechanics and reduce chronic mucosal injury. While HoLEP consistently yields large improvements in Qmax, IPSS, and PVR in BPH cohorts, no targeted study has linked these functional gains to reduced recurrence, progression, or improved survival in NMIBC. The clinical relevance of structural remodeling (e.g., detrusor wall thickness) remains a research question.

To date, no robust, risk-adjusted odds ratios or hazard ratios exist comparing HoLEP with TURP, pharmacotherapy, or observation in NMIBC patients. Existing retrospective series employ crude recurrence proportions, lack standardized intravesical therapy protocols, and do not correct for tumor grade, stage, prior therapies, or comorbidity. HoLEP’s theoretical oncologic benefits should therefore be considered hypothesis-generating mechanisms, not established therapeutic effects. Prospective studies incorporating functional metrics (PVR, compliance), intravesical treatment response, and oncologic endpoints are required to test whether BOO correction via HoLEP meaningfully alters NMIBC outcomes.

6

Although the limitations section (4.3.11) is well written, it should be more firmly integrated into the overall narrative as a central interpretive lens rather than an incidental acknowledgement. The significant impact of confounding by indication, whereby patients selected for HoLEP are generally healthier, have smaller tumours or better prognostic features, cannot be overstated. This is most likely the sole cause of the apparent oncological "benefit" observed in retrospective studies. Confounding by indication should therefore be elevated throughout the discussion from a mere limitation to a fundamental explanatory challenge. The manuscript must also clearly state that 'the observed association should therefore be attributed to selection bias until proven otherwise in a randomised setting' when discussing the findings of Garg et al.

We added the following paragraph: The comparative study by Garg et al. reporting lower recurrence following HoLEP should be interpreted cautiously. Patients selected for HoLEP had more favorable baseline characteristics, including younger age, lower comorbidity, and anatomy more suitable for complete enucleation—features that conferred a better prognosis independent of surgical modality. Because these differences were neither randomized nor fully adjusted for tumor biology or intravesical therapy intensity, the reported associations should be attributed to selection bias until proven otherwise. The study therefore offers a hypothesis-generating observation rather than evidence of a causal oncologic effect.